# KLB dysregulation mediates disrupted muscle development in intrauterine growth restriction

Yennifer Cortes-Araya , Claire Stenhouse, Mazdak Salavati, Susan O. Dan-Jumbo, William Ho, Cheryl J. Ashworth, Emily Clark, Cristina L. Esteves and F. Xavier Donadeu

*Division of Functional Genetics and Development, The Roslin Institute and Royal (Dick) School of Veterinary Studies, University of Edinburgh, Edinburgh, UK*

Edited by: Laura Bennet & Janna Morrison

The peer review history is available in the Supporting Information section of this article (https://doi.org/10.1113/JP281647#support-information-section).

**Abstract**   Intrauterine growth restriction (IUGR) is a leading cause of neonatal morbidity and mortality in humans and domestic animals. Developmental adaptations of skeletal muscle in IUGR lead to increased risk of premature muscle loss and metabolic disease in later life. Here, we

identified $\beta$-Klotho (KLB), a fibroblast growth factor 21 (FGF21) co-receptor, as a novel regulator of muscle development in IUGR. Using the pig as a naturally-occurring disease model, we performed transcriptome-wide profiling of fetal muscle (day 90 of pregnancy) from IUGR and normal-weight (NW) littermates. We found that, alongside large-scale transcriptional changes comprising multiple developmental, tissue injury and metabolic gene pathways, KLB was increased in IUGR muscle. Moreover, FGF21 concentrations were increased in plasma in IUGR fetuses. Using cultures of fetal muscle progenitor cells (MPCs), we showed reduced myogenic capacity of IUGR compared to NW muscle *in vitro*, as evidenced by differences in fusion indices and myogenic transcript levels, as well as mechanistic target of rapamycin (mTOR) activity. Moreover, transfection of MPCs with KLB small interfering RNA promoted myogenesis and mTOR activation, whereas treatment with FGF21 had opposite and dose-dependent effects in porcine and also in human fetal MPCs. In conclusion, our results identify KLB as a novel and potentially critical mediator of impaired muscle development in IUGR, through conserved mechanisms in pigs and humans. Our data shed new light onto the pathogenesis of IUGR, a significant cause of lifelong ill-health in humans and animals.

(Received 20 November 2021; accepted after revision 4 January 2022; first published online 26 January 2022)

**Corresponding author** F. Xavier Donadeu: Division of Functional Genetics and Development, The Roslin Institute and Royal (Dick) School of Veterinary Studies, University of Edinburgh, Easter Bush, Midlothian, EH25 9RG Scotland, UK. Email: xavier.donadeu@roslin.ed.ac.uk

**Abstract figure legend** In intrauterine growth restriction (IUGR) fetuses, reduced placental supply induces an adaptive response characterized by preferential shunting of blood and therefore oxygen and nutrients to vital tissues, such as brain and heart, at the expense of other tissues, including skeletal muscle. Using a pig model, we found skeletal muscle from IUGR fetuses to display large-scale gene expression dysregulation, including developmental, tissue injury and metabolic genes. Among the up-regulated genes in IUGR muscle was the fibroblast growth factor 21 (FGF21) co-receptor, $\beta$-Klotho (KLB), whereas FGF21 levels were distinctly elevated in the circulation of IUGR fetuses. Subsequent studies with muscle progenitor cells showed that signalling through FGF21 and KLB inhibits mechanistic target of rapamycin activation and reduces differentiation and myotube formation by both pig and human cells. These results identify FGF21/KLB signalling as a novel mediator of reduced muscle growth in IUGR fetuses.

## Key points

- Intrauterine growth restriction (IUGR) is associated with large-scale transcriptional changes in developmental, tissue injury and metabolic gene pathways in fetal skeletal muscle.
- Levels of the fibroblast growth factor 21 (FGF21) co-receptor, $\beta$-Klotho (KLB) are increased in IUGR fetal muscle, and FGF21 concentrations are increased in IUGR fetal plasma.
- KLB mediates a reduction in muscle development through inhibition of mechanistic target of rapamycin signalling.
- These effects of KLB on muscle cells are conserved in pig and human, suggesting a vital role of this protein in the regulation of muscle development and function in mammals.

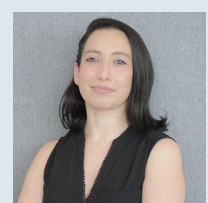

**Yennifer Cortes Araya** obtained her DVM from the University of Chile and her MSc in Animal Bioscience from the University of Edinburgh. She recently completed her doctoral thesis on the effects of intrauterine growth restriction (IUGR) on porcine muscle development under the guidance of Dr. Xavier Donadeu at the Division of Functional Genetic and Development of The Roslin Institute, University of Edinburgh. During these years her research interest has been in the area of Stem Cell biology, early life 'programming' and impacts upon foetal growth and development in large animals. She addresses these topics from a 'One Health' perspective, focusing her research on using farm animals as research models to be used in translational medicine.

## Introduction

Intrauterine growth restriction (IUGR) affects ∼5−8% of all human births worldwide and is a leading cause of neonatal morbidity and mortality (Clark *et al.* 2020). IUGR arises when placental nutrient supplies fail to satisfy the requirements of the developing fetus. IUGR babies typically present with low birth weight and morphological features from altered allometric organ growth, and are prone to perinatal complications affecting multiple body systems. Critically, IUGR individuals are also at increased risk for a myriad of diseases later in life, including metabolic, cardiovascular, renal, hepatic, ovarian and neurological/cognitive disorders (Brown & Hay, 2016; Sharma *et al.* 2016). Because harvesting human fetal tissues is often impractical, most knowledge on IUGR pathophysiology comes from studies in animal models, particularly large animals such as sheep and pigs in which the physiology most closely resembles that of humans (Swanson & David, 2015). Being a litter-bearing species, IUGR occurs naturally in the pig. As in most cases of human IUGR, IUGR in the pig results from placental insufficiency, which, in this particular species, arises as a consequence of uterine crowding in highly prolific breeds (Foxcroft *et al.* 2006). Thus, the pig provides a particularly convenient model for investigating the developmental pathophysiology of IUGR.

In IUGR fetuses, reduced placental supply induces an adaptive response that preferentially shunts oxygen and nutrients to vital tissues; namely, the brain and heart. As a consequence, available resources for muscle growth are significantly reduced, resulting in a reduction in the number of muscle fibres at birth. This phenotype cannot be fully compensated for by postnatal growth, thus resulting in a permanent reduction in total muscle. This is associated with life-long impairment in muscle function and a predisposition to diseases such as sarcopenia, obesity and diabetes (Brown, 2014). Moreover, in livestock species such as pigs, IUGR is associated with a significant reduction in meat production and quality, as well as being an important financial and animal welfare problem for that key livestock industry (Bérard *et al.* 2008). Thus, elucidating the basic mechanisms underlying impaired skeletal muscle development is important for understanding how IUGR contributes to long-term health and disease, and may provide strategies to improve productivity in livestock.

Profound adaptive changes in skeletal muscle metabolism occur in response to reduced nutrient availability in IUGR. The results of numerous studies in sheep and rodents (Brown & Hay, 2016; Chen *et al.* 2017a; Stremming *et al.* 2020) indicate that, amongst other changes, fetal muscle adapts to IUGR by reducing mitochondrial oxidative phosphorylation capacity and glucose oxidation, at the same time as increasing fatty acid and amino acid oxidation as sources of fuel. Furthermore, amino acid uptake and protein accretion rates in hind limbs of IUGR sheep fetuses were reduced (Rozance *et al.* 2018; Stremming *et al.* 2020), whereas both a reduction in protein accretion (Chen *et al.* 2017a) and an increase in protein degradation (Wang *et al.* 2008) were reported in the muscle of IUGR piglets. Increased adrenergic activity triggered by hypoxemia and reduced nutrient availability appears to mediate metabolic adaptations in IUGR fetuses. This involves a catecholamine-induced reduction in pancreatic insulin production and impaired insulin signalling impacting on Akt-mechanistic target of rapamycin (mTOR) activity in muscle cells (Brown & Hay, 2016; Limesand & Rozance, 2017). Associated with these effects is an increase in insulin-like growth factor binding protein 1 secretion and activity, which leads to decreased insulin-like growth factor 1 availability (Damerill *et al.* 2016), in turn resulting in reduced skeletal muscle growth. As a result, nutrient requirements for skeletal muscle development decrease in the IUGR fetus. Accordingly, myoblasts from offspring of nutrient-restricted ewes (Yates *et al.* 2014a; Soto *et al.* 2017) or low birth weight piglets (Nissen & Oksbjerg, 2009) displayed a reduced capacity to proliferate and/or form myotubes *in vitro*. These findings suggest developmental programming of muscle progenitor cells by IUGR, although the mechanisms involved have not been elucidated.

An adequate understanding of the mechanisms driving impaired myogenesis in IUGR is required for developing effective strategies to ameliorate its detrimental effects on life-long health in humans and animals. To that end, elucidation at the genome-wide level of the response of skeletal muscle to IUGR in the developing fetus will be highly valuable. To our knowledge, only one study to date has undertaken large scale gene profiling of fetal IUGR muscle (Soto *et al.* 2017). In that study, muscle samples from a sheep model of temperature-induced placental insufficiency were analysed using a bovine microarray platform, focusing mainly on differentially expressed cell cycle genes. In the present study, we undertook an unbiased genome-wide approach using RNA sequencing to identify global gene expression signatures in fetal skeletal muscle from pigs, increasingly recognized as a high value animal model of IUGR (Che *et al.* 2010; Ebner *et al.* 2014; Boubred *et al.* 2017; Bæk *et al.* 2019; Gao *et al.* 2020). In addition to numerous anatomic, metabolic and genetic similarities with humans, a distinct advantage of the pig is that comparisons between IUGR and normal weight littermates can be made, thus avoiding confounding effects of genetic background or maternal factors. Using this model, we found gene expression in IUGR muscle to be widely dysregulated, including numerous developmental, tissue injury and metabolic gene pathways. Following these analyses, we tested the hypothesis that $\beta$-Klotho (KLB), a fibroblast growth factor

21 (FGF21) co-receptor that is up-regulated in IUGR littermates, mediates at least some of the deleterious effects of IUGR on muscle development.

## Methods

### Ethical approval

All animal procedures were performed with approval from The Roslin Institute (University of Edinburgh) Animal Welfare and Ethical Review Board and following the UK Animals (Scientific Procedures) Act, 1986.

For human samples, written maternal informed consent in compliance with the the *Declaration of Helsinki* was obtained, and the study was approved by the Lothian Research Ethics Committee in Scotland (ref 08/S1101/1). All methods were performed following the relevant guidelines and regulations of this approval.

### Sample collection

Ten Large White × Landrace gilts aged 11−14 months were held at the University of Edinburgh's Dryden Farm Large Animal Unit under a commercial cob-based diet formulation and free access to water. Gilts were inseminated with semen from three Large White sires and killed using sodium pentobarbitone (Henry Schein Animal Health, Dumfries, UK; 20% w/v, 0.4 mL kg$^{-1}$ by ɪ.ᴠ. injection via a cannula inserted in the ear vein) on day 90 of pregnancy (pregnancy length, ∼115 days); this corresponds to a stage when hyperplasic muscle fibre development (primary and secondary fibres) has just been completed (Wigmore & Stickland, 1983). After death, the uterus was quickly dissected, and all fetuses were removed, weighed and visually sexed. IUGR fetuses were defined as having a weight >2 SD below the average litter weight. From those litters containing an IUGR male, the IUGR and two normal weight (NW) male littermates were selected. The two NW fetuses from each litter were chosen among non-IUGR littermates with body weights above and below, respectively, the litter NW average. The two fetuses were respectively assigned to NW sets 1 and 2. Set 1 was used for the majority of subsequent analyses, whereas set 2 was used only as an additional control for quantitative real-time PCR (qPCR) validation of the sequencing results, as described below. Immediately after uterine dissection, body and organ measurements were taken from each fetus. Samples of semi-tendinosus muscle were also taken and snap-frozen in liquid nitrogen, embedded in OCT and snap-frozen or, alternatively, transported to the laboratory in ice and digested for cell culture, as described below. Blood samples were also collected by cardiac puncture. Plasma was harvested by centrifugation and stored at −20°C. In addition, five male piglets (NW,

from separate litters) were killed at birth, and blood and muscle samples were collected as described above.

Human fetal hind limb muscle ($n = 3$, 10−20 weeks of gestation) was obtained following medical termination of pregnancy at the Simpson Centre for Reproductive Health, Royal Infirmary of Edinburgh, UK, and gestational ages were determined as described previously (Hartanti *et al.* 2020). Pregnancies were all terminated for social reasons, and all fetuses appeared morphologically normal.

### RNA sequencing and data analysis

Muscle samples (30 mg) from porcine fetal pairs (IUGR and NW, $n = 4$ litters) were homogenized in RNABee (AMS Biotechnology, Abingdon, UK) in Lysing Matrix D tubes (MP Biomedicals, Illkirch, France) and extracted in accordance with the manufacturer's instructions, followed by transfer to a RNeasy Mini Spin column and treatment with RNase-free DNase (Qiagen, Manchester, UK). RNA concentration and quality, as defined by the RNA Integrity Number equivalent (RINe), were determined by Tapestation 2200 (Agilent Technologies, Edinburgh, UK). All samples used for sequencing had RINe values >8.5. RNA libraries were prepared by Exiqon A/S (Vedbæk, Denmark) using an Illumina TruSeq Stranded mRNA Library Prep Kit (Illumina, San Diego, CA, USA) and sent to 50 bp/30 M read, single-end sequencing using the Illumina HiSeq2500 platform. After intensity correction and base calling, FASTQ files were generated using bcl2fastq software (Illumina), including quality scoring of each individual base. Genes were identified by alignment to the reference transcriptome. Briefly, the raw RNA sequencing data was trimmed using Trimmomatic, v0.39.0 (*SLIDINGWINDOW:5:20 MINLEN:30*) (Bolger *et al.* 2014) and aligned using Kallisto, v0.43.0 (Bray *et al.* 2016) to the cDNA level transcriptome assembly of *Sscroffa11.1* (ftp://ftp.ensembl.org/pub/release-100/fasta/sus_scrofa/ cdna/Sus_scrofa.Sscrofa11.1.cdna.all.fa.gz). TPM (transcript per kilo base million) counts were used for downstream analysis in R (tximport v1.18.0 and DESeq2 v1.30.0; apeglm shrinkage model) (Love *et al.* 2014) and differentially expressed genes were identified between IUGR and NW littermates accounting for effects of litter. Differentially expressed genes were analysed using Ingenuity Pathway Analysis software (Qiagen) to identify (from the Qiagen Knowledge Base) significantly over-represented biological pathways using right-tailed Fisher's exact tests ($P < 0.01$). Because, after false discovery rate (FDR) adjustment, a relatively low number of genes remained differentially expressed between NW and IUGR littermates, to maximize gene representation, all genes differentially expressed

($P < 0.05$) before adjustment were included in the Ingenuity Pathway Analysis analyses. Raw sequencing data files (FASTQ) were deposited in NCBI BioProject database (https://www.ncbi.nlm.nih.gov/bioproject) under accession number PRJNA678714.

## qPCR

Total RNA (1$\mu$g) from muscle tissue or cells was reverse transcribed using Superscript III (Thermo Fisher Scientific, Waltham, MA, USA) and a Whatman-Biometra Thermocycler (Biometra, Göttingen, Germany). RNA was mixed with 1 $\mu$L of Random Primers (Promega, Madison, WI, USA), 1 $\mu$L of dNTP mix (Invitrogen, Carlasbad, CA, USA) and nuclease-free water up to 13 $\mu$L in a 200 $\mu$L nuclease-free microcentrifuge tube. Samples were heated to 65°C for 5 min and then placed at 4°C for 5 min. Tubes were then centrifuged briefly, and 4 $\mu$L of 5X First-Strand Buffer, 1 $\mu$L of 0.1 DTT, 1 $\mu$L of RNasin Plus Rnase Inhibitor (#N2611; Promega) and 1 $\mu$L of SuperScript III were added and mixed by pipetting. The samples were then heated to 25°C for 5 min, 50°C for 1 h and 70°C for 15 min, after which they were used immediately for qPCR or were frozen at −20°C. qPCR was performed as described previously (Weatherall *et al.* 2020), using Sensi-FAST SYBR Lo-ROX (Bioline, London, UK) and validated species-specific primers (see Supporting information, Table S1) in an MX3005P system (Stratagene, La Jolla, CA, USA) and data were analysed with MxPro (Agilent Technologies). Primers were validated by confirming amplification efficiencies of 90−110% using a standard curve (using sequential 1:4 dilutions of a 1:8 cDNA dilution) prepared using skeletal muscle or pooled cell samples, as well as by the presence of a single peak in the reaction's dissociation curve. For each specific transcript analysed, a sample dilution was subsequently used that yielded $C_t$ values in the middle of the linear portion of the standard curve. Expression levels for each transcript were determined relative to the above standard curve, and normalized to levels of the stable genes: 18S, TOP2B, RPL4 and HPRT1.

## Immunochemistry

Tissue cryosections (10 $\mu$m) were stained with primary antibody (see Supporting information, Table S2) at 4°C overnight, washed with phosphate-buffered saline (PBS), and incubated with the respective secondary antibody (see Supporting information, Table S2) for 1 h at room temperature. Slides were then washed and mounted in Fluoroshield with 4′,6-diamidino-2-phenylindole (DAPI) (Sigma-Aldrich, St Louis, MO, USA). Three images from each of two tissue sample sections were taken using a DMLB fluorescence microscope (Leica,

Wetzlar, Germany). In total, ∼100 muscle fibres were analysed from each fetus. Intensities from secondary antibody-stained control sections were used for background normalization in each case.

Cells were fixed and permeabilized in ice-cold methanol: acetone (50:50) for 10 min at room temperature, followed by washing with PBS for 5 min and incubation with protein block solution (Springbio, Farnborough, UK) for 1 h at room temperature. Cells were stained with primary antibody at 4°C overnight, washed with PBS and then incubated with the respective secondary antibody for 1 h at room temperature in the dark, before being washed and mounted in Fluoroshield with DAPI (Sigma-Aldrich), sealed with a coverslip and examined using an Axiovert 25 (Zeiss, Oberkochen, Germany) inverted fluorescence microscope. Three images were taken from duplicate wells using an Axiocam 503 high-resolution colour camera/Zen software (Zeiss). Fusion index (i.e. the ratio between the number of nuclei within myotubes and the total number of nuclei per field) was determined from myosin heavy chain (MYHC) stained pictures using ImageJ (NIH, Bethesda, MD, USA). In all cases, intensities from secondary antibody-stained control wells were used for background normalization.

## Plasma FGF21 and FGF19 quantification

FGF21 and FGF19 concentrations were determined in duplicate plasma samples using an enzyme-linked immunosorbent assay kits, EP0057 (FineTest Biotech, Wuhan, China) and ab273220 (Abcam, Cambridge, UK), respectively, in accordance with the manufacturer's instructions. Intra-assay coefficient of variation and assay sensitivity were 3.8% and 18.75 pg mL$^{-1}$ for FGF21, and 8.0% and 10.0 pg mL$^{-1}$ for FGF19.

## Primary muscle progenitor cell (MPC) isolation, culture and differentiation

Progenitor-enriched cell populations were isolated from muscle samples as described previously (Vaughan & Lamia, 2019), and cultured on Matrigel (BD Biosciences, Franklin Lake, NJ, USA) at 39°C in Dulbecco's modified Eagle's medium (DMEM) high glucose with 1% P/S, supplemented with 20% fetal bovine serum (FBS) (Life Technologies, Carlsbad, CA, USA) and 5 ng mL$^{-1}$ basic FGF (PeproTech, London, UK). Cells were trypsinized and passaged every 2−3 days. MPCs were differentiated using a protocol adapted from Hausman & Poulos (2005). In short, MPCs were plated on rh-Laminin 521 (Life Technologies)-coated wells (1000 cells mm$^{-2}$) and, when they reached 70% confluency, media was changed to DMEM high glucose with antibiotics supplemented with 10% FBS and 80 nm dexamethasone (Sigma-Aldrich).

**Table 1. Body and organ measurements (mean ± SD) from NW and IUGR littermates used for analyses**

| Variable | NW | NW* | IUGR |
|---|---|---|---|
| Fetal weight (g) | 761.94 ± 79.55[a] | 682.09 ± 130.36[a] | 446.22 ± 102.68[b] |
| Crown–rump length (mm) | 245.67 ± 30.92 | 257.00 ± 28.86* | 216.80 ± 31.45 |
| Brain weight (g) | 19.19 ± 3.69 | 17.44 ± 1.02 | 17.84 ± 1.16 |
| Liver weight (g) | 14.70 ± 4.83 | 14.06 ± 1.55 | 12.05 ± 2.27 |
| Brain (% body weight) | 2.34 ± 0.16[a] | 2.59 ± 0.45[a] | 4.01 ± 0.83[b] |
| Brain to liver weight ratio | 1.36 ± 0.32 | 1.26 ± 0.19* | 1.52 ± 0.32 |

Values for each of two different sets of NW littermates, NW (used for sequencing, $n = 3$ litters) and NW* (used as an additional control for PCR validation of sequencing data, $n = 5$ litters), as well as IUGR littermates ($n = 5$ litters), are shown separately (values do not include data from the outlier litter). Mean ± SD litter size = 14.0 ± 2.2 fetuses, mean ± SD fetal weight = 705 ± 95.9 g ($n = 5$ litters). Means with different superscript letters within an end-point are different ($P < 0.05$). An asterisk indicates a significant difference ($P < 0.05$) between NW* and IUGR littermates only.

Forty-eight hours later or when cells reached full confluence (day 0), media was changed to DMEM high glucose with antibiotics supplemented with 2% FBS, 1% of insulin-transferrin-selenium (Life Technologies) and, in some experiments, human FGF21 ($1-100$ ng mL$^{-1}$; #100-42; PeproTech), and maintained for up to 7 days. All experiments with cells were performed using triplicate wells.

### RNA interference

On day $-1$ (i.e. when cells had typically reached $50-60\%$ confluency), MPCs were transfected with two small interfering RNAs (siRNAs) targeting porcine KLB (5′-GAACCAAACAGAUCAGAAAUU-3′ and 5′-CGUUGGAACUGGAGCAUUUUU-3′, 25 nm each; Dharmacon, Cambridge, UK) or a scrambled RNA sequence (control siRNA; 50 nm) using Hiperfect reagent (Qiagen), in accordance with the manufacturer's instructions.

### Western blotting

Total protein was extracted from fully confluent 12-well plates by adding RIPA lysis buffer with Halt Protease phosphatase inhibitor (#78440; Invitrogen). Protein (50 $\mu$g) was diluted in 2× Laemmli sample buffer (dilution 1:1; #161-0737; Bio-Rad, Hercules, CA, USA) and 2-mercaptoethanol (355 mm; #161-0710; Bio-Rad) and heated for 5 min at 95°C, then electrophoresed in $4-20\%$ Mini-PROTEAN TGX Precast Protein gels (Bio-Rad) in a Mini Trans-Blot Cell (Bio-Rad) at 150 V for 90 min. Gels were transferred to PVDF membrane iBlot Transfer Stacks (#IB24001; Thermo Fisher Scientific) using programme three of an iBlot Transfer (#IB21001; Thermo Fisher Scientific). After blocking with Intercept (TBS) Blocking buffer (#927-60001; LI-COR Biosciences,

Lincoln, NE, USA) for 1 h at room temperature, membranes were incubated with primary antibody (see Supporting information, Table S2) overnight at 4°C, followed by washing and incubation with a secondary 680RD antibody for 1 h and visualization with a LI-COR Odyssey IR imaging scanner. Signal intensities were quantified using Image Studio Lite 5.0 (LI-COR).

### Statistical analysis

All statistical analyses were performed using Mini-tab, version 18 (Minitab Inc., State College, PA, USA). Data were assessed for normality using the Kolmogorov–Smirnoff test ($P > 0.01$) and log-transformed before analyses if needed. Outlier data points identified using Grubb's test were excluded. Data were then analysed using one- or two-way ANOVA with litter as covariate followed by a *post hoc* Tukey's test or, if only two means were compared, Student's *t* tests. $P < 0.05$ was considered statistically significant.

### Results

#### Fetal IUGR muscle displays wide transcriptional dysregulation, including numerous pathways involved in development, tissue injury and metabolism

Out of the 10 pig litters used in the present study, seven contained a single IUGR fetus (five male and two female) and one contained two IUGR fetuses (male and female), as defined by a weight >2SD below the average litter weight. Thus, for the experimental analyses described below, we used male NW and IUGR littermates (selected as described in the Methods in 'Sample collection') from the six litters containing an IUGR male. Fetuses classed as IUGR had a higher mean brain weight as a percentage

of body weight than NW littermates (Table 1), confirming their growth-restricted status. Moreover, IUGR muscle contained thinner fibres and higher fibre densities than NW muscle (Fig. 1), consistent with previous findings (Wigmore & Stickland, 1983; Felicioni *et al.* 2020; Stange *et al.* 2020).

To identify transcriptome-wide signatures in IUGR skeletal muscle, we performed RNA sequencing on paired samples from IUGR and NW fetuses from four different litters. RNA sequencing produced high-quality data from all samples, as determined by *Q* scores >30 for both read quality and base quality. A mean ± SD of 54.2 ± 1.6 million reads was obtained per sample, 79.9 ± 0.28% of which mapped to a total of 17,600 ± 29.4 genes in the reference porcine genome (see Supporting information, Table S3). Principal component analysis on the 500 genes with the largest coefficient of variation identified the IUGR sample from one litter to be an outlier (litter 4) (Fig. 2*A*). Incidentally, this litter was distinct in that it was the only one that contained two IUGR fetuses, one of each sex, of which the female was the lightest. This litter was removed from all subsequent analyses.

In total, 1031 differentially expressed genes were identified (*P* < 0.05) between IUGR and NW fetuses (Fig. 2*B*; see also Supporting information, Tables S4 and S5). After FDR adjustment (FDR < 0.1), 38 and 43 genes were up-regulated and down-regulated, respectively, in IUGR relative to NW fetuses. Gene

Ontology (http://geneontology.org) analysis of all differentially expressed genes revealed significant enrichment (*P* < 0.01) for terms broadly related to Development, Tissue Injury and Metabolism (Fig. 3; see also Supporting information, Table S6). 'Development' included pathways involved in skeletal muscle and neural development, with IUGR muscle displaying down-regulated levels of several related genes, including MYOG (a myogenic transcription factor), NKX62 (a gene involved in somatic motor neuron development) (Pattyn *et al.* 2003), RET and ACTN3 (Fig. 4; see also Supporting information, Table S5). In that regard, dysregulated neurodevelopment is a well described feature of IUGR (Wixey *et al.* 2019; Mallard *et al.* 2000), although this has not been reported in the context of skeletal muscle. Tissue injury pathways could be classified into those associated with Inflammation, Coagulation and Anti-oxidation/Detoxification. Numerous transcripts corresponding to those categories were highly up-regulated in IUGR fetal muscle (Fig. 4; see also Supporting information, Table S4), particularly those related not only to coagulation (e.g. PLG, SERPINA5, F5, F9, ITIH2, FGG), but also to inflammation (e.g. AMBP, CCL16) and detoxification (e.g. ABCC6). The largest functional category was Metabolism. It included several signalling pathways and transcripts broadly involved in the regulation of metabolism (Fig. 3), many of which were up-regulated in IUGR muscle such as IGBP1, AHSG and KLB. It also included specific metabolic pathways, from which multiple transcripts were highly up-regulated in IUGR muscle (Fig. 4; see also Supporting information, Table S4), including glucose metabolism (ALDOB), lipid biosynthesis and transport (APOC2, APOB, CIDEB, PNPL) and amino acid degradation (TAT, TDO2, PRODH2). Finally, several entities under the ontology category, Disease, were also enriched and corresponded to later life metabolic and other diseases often associated with IUGR, including endocrine (diabetes), as well as hepatic and vascular disorders (see Supporting information, Table S6).

The results of RNA sequencing were validated by qPCR for a selected group of genes (see Supporting information, Tables S4 and S5) using an extended group of samples, including an additional set of NW littermates as described in the Methods for 'Sample selection'. Genes for validation were chosen that (1) represented the three functional categories above, (2) had well-defined biological function(s) and (3) were detectable by qPCR in a majority of the samples analysed. As shown in Fig. 4, differences in expression levels detected between IUGR and NW fetuses were highly consistent between the two analytical methods compared, as well as between the two sets of NW littermates when compared with IUGR fetuses from the same litters (indicated by PCR and PCR* in Fig. 4).

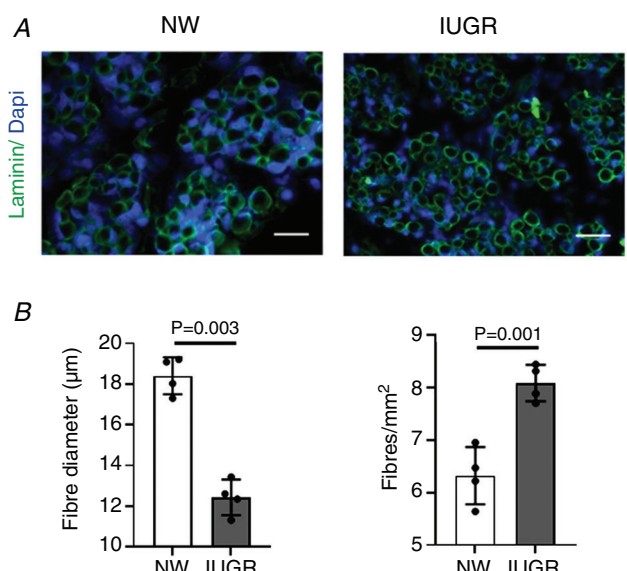

**Figure 1. Histological features of skeletal muscle from porcine fetal NW and IUGR littermates**

*A*, representative images of sections of semi-tendinosus muscle from NW and IUGR littermates that were immunostained for laminin (green) and counterstained with nuclear marker, DAPI (blue). *B*, diameter and density of muscle fibres (mean ± SD together with individual data points) in the two groups. Scale bars = 100 μm. [Colour figure can be viewed at wileyonlinelibrary.com]

## Reduced muscle development in IUGR littermates is associated with increased KLB levels compared to NW littermates

As indicated above, KLB was among the up-regulated transcripts in IUGR littermates. KLB is an obligatory co-receptor of the metabolic hormone, FGF21. Given the proposed role, through binding to FGF21, as a master regulator of the starvation response (Inagaki *et al.* 2007; Tyynismaa *et al.* 2010), we focused our subsequent attention on KLB. The role of KLB in regulation of energy metabolism in adipose tissue and liver has been reported

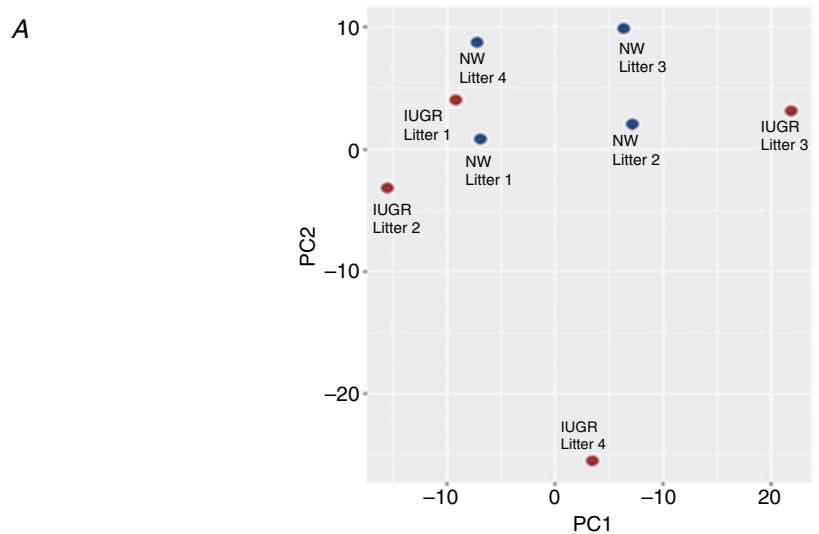

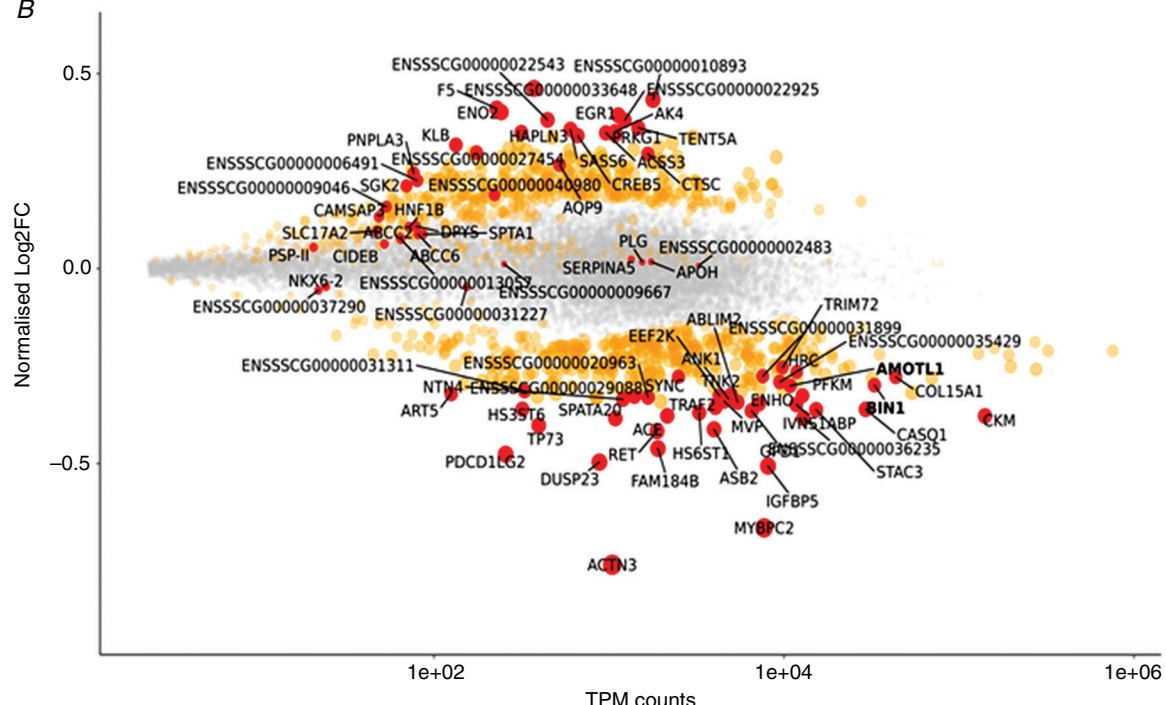

**Figure 2. Results of RNA sequencing of muscle samples from porcine NW and IUGR littermates**
*A*, results from principal component (PC) analysis on the 500 genes with the largest coefficient of variation based on normalized read counts. Each circle represents a sample. *B*, scatter plot representation of all mapped genes according to their abundance (normalized log2 fold change against TPM counts) in IUGR relative to NW muscle. Genes differentially expressed in IUGR muscle (*P* < 0.05) are shown in orange (*n* = 3 litters). Genes differentially expressed after FDR adjustment (<0.1) are shown in red. TPM, transcripts per kilo base million = read counts divided by the length of each gene in kilobases. [Colour figure can be viewed at wileyonlinelibrary.com]

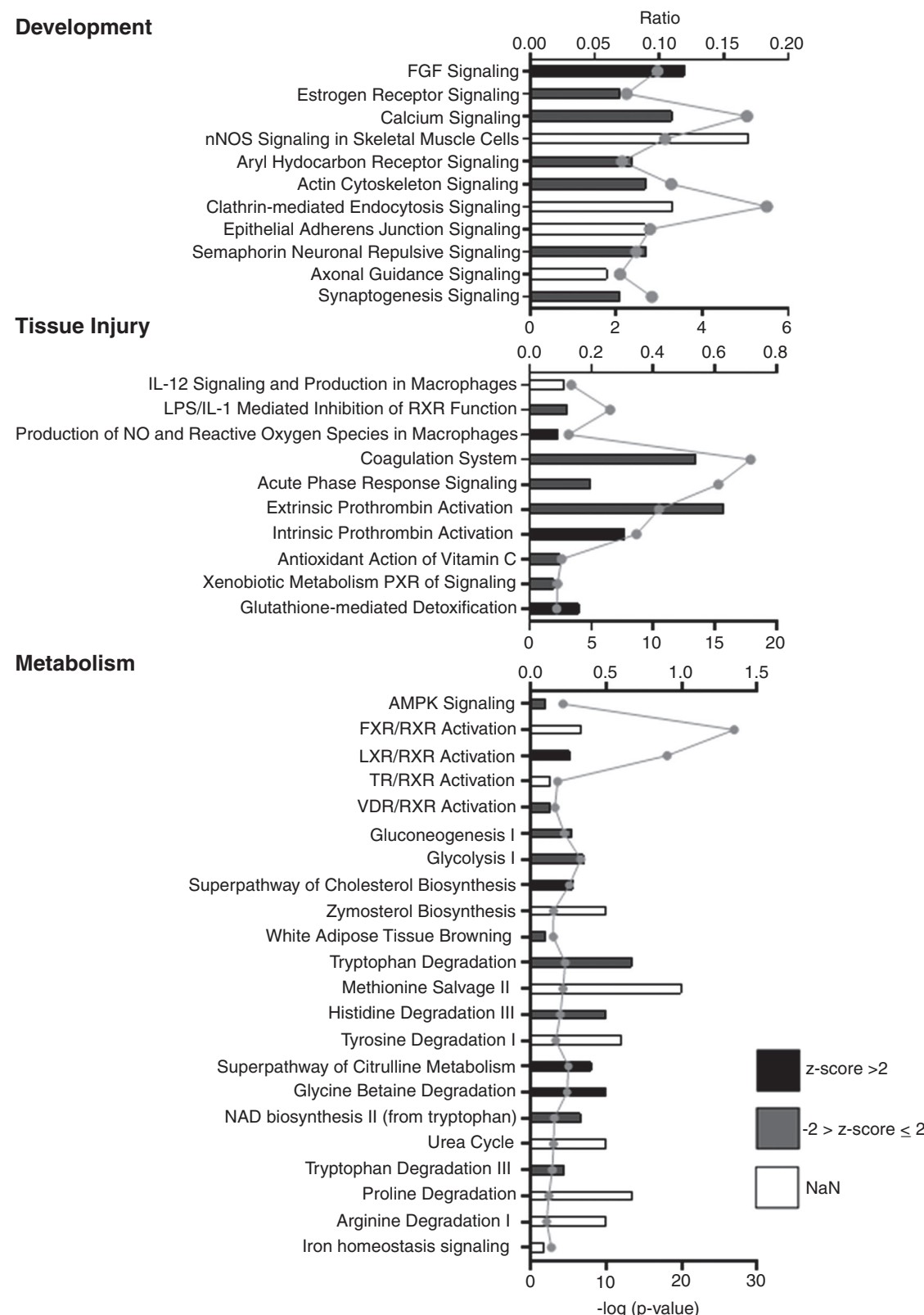

**Figure 3. Results of Gene Ontology analysis on differentially expressed genes identified by RNA sequencing**

Biological pathways found to be over-represented ($P < 0.01$) among differentially expressed genes could be classified into Metabolic, Development and Tissue Injury categories. In each panel, 'ratio' (represented by bar

size) corresponds to the number of genes that map to a specific pathway divided by the total number of genes in the pathway. The statistic '*z* score' (represented by column colour) is used to infer the likely activation state of each pathway within the specific biological context under consideration based on comparison with a model assigning random regulation directions. A *z* score >2 or <−2 is considered as a strong indicator that a pathway is up-regulated or down-regulated, respectively. NaN, undetermined. −Log (*P* values) are represented by the grey dotted line.

previously (Kurosu *et al.* 2007), although little is known about its effects on muscle function and how it mediates tissue responses to starvation in the developing fetus. To confirm the results of RNA analyses, we first performed immunofluorescence and showed that KLB is indeed present in porcine fetal muscle (Fig. 5*A*), in agreement with

results obtained in other species (Benoit *et al.* 2017). Moreover, we found that the mean levels of KLB protein were around three times higher in IUGR than NW fetuses, consistent with the results of qPCR (Fig. 5*B*).

We then aimed to determine whether differences in myogenic capacity could be detected in cultured cells.

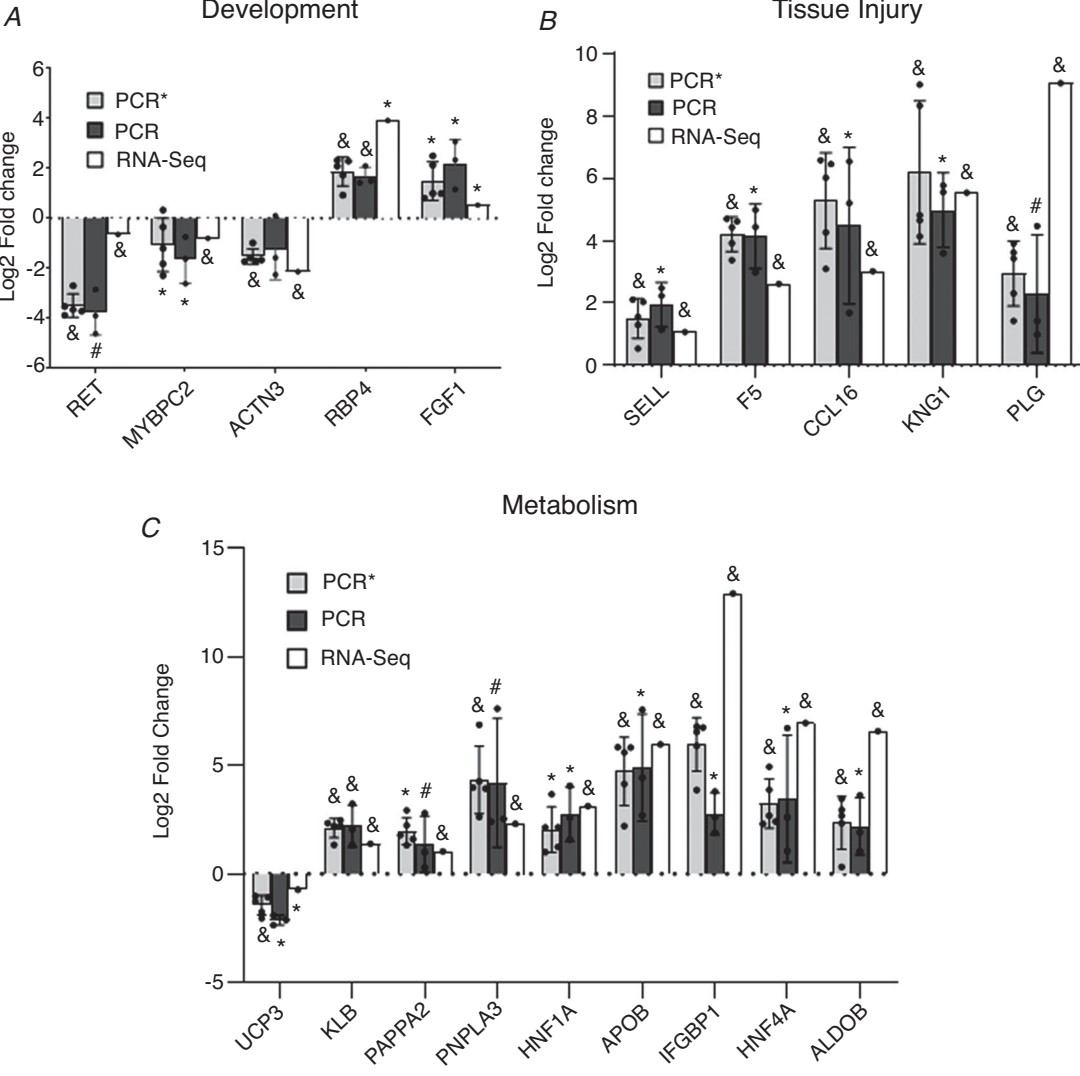

**Figure 4. Validation of RNA sequencing results by qPCR**
Comparison of log2 fold-change expression values (IUGR/NW) for selected transcripts related to development *(A)* tissue Injury *(B)* and metabolism *(C)* obtained by RNA sequencing (RNA-seq) or qPCR (PCR) of fetal muscle samples from three pig litters. Additional qPCR validation was performed (indicated as PCR*) using a different set of NW samples (and litter-matched IUGR samples) from a total of five litters. In all cases, signficant gene expression up-regulation or down-regulation in IUGR relative to NW littermates is indicated by an asterisk (*) (*P* < 0.05) or an ampersand (&) (*P* < 0.01), whereas differences approaching signficance (*P* < 0.1) are indicated by a hash (#) symbol. Mean ± SD values are shown together with individual data points.

We found that MPCs from IUGR fetuses had reduced myogenic capacity in culture compared to cells from NW littermates (Fig. 6A), consistent with previous results obtained using muscle cells from sheep or pig IUGR offspring (Yates *et al.* 2014b; Chen *et al.* 2017b). This was confirmed by differences in fusion indices (Fig. 6B), as well as in transcript levels of both the developmental myosin, MYH3, and the transcriptional factor involved in terminal myoblast differentiation, MYOG (Fig. 6C). We then examined mTORC1 activity, a primary driver of muscle growth (Ge & Chen, 2012), and found that mean phosphorylation levels of both mTOR at Ser2448, and its effector, S6K1, at Thr389, were lower in IUGR myotubes, although these differences did not reach significance ($P > 0.05$) (Fig. 6D). Finally, in line with the data from muscle tissues (Fig. 5), *in vitro* derived myotubes expressed KLB. Moreover, KLB protein and mRNA were expressed at higher levels in IUGR-derived compared to NW-derived muscle cells (Fig. 6E and F).

### KLB knockdown promotes myogenesis and mTOR activation in MPCs

To investigate whether the increased levels of KLB may indeed result in attenuated muscle development in

IUGR pigs, we first aimed to determine the effects of KLB activation with FGF21 or down-regulation with siRNAs on the myogenic capacity of muscle cells using NW-derived MPCs. To this end, MPCs were first transfected with KLB siRNAs 24 h before inducing myogenesis (Fig. 7A–C). KLB down-regulation was associated with an increase in myogenesis (Fig. 7D), with siRNA-treated cells displaying higher fusion indices (Fig. 7E) and higher levels of myogenic markers (Fig. 7F) than control cells upon induced differentiation. Moreover, phosphorylation of S6K1 increased in siRNA-treated cells (Fig. 7G), suggesting that the effects of KLB on myogenesis occur, at least in part, through inhibition of mTORC1 signalling.

### Treatment with the KLB ligand, FGF21, inhibits myogenesis and mTOR activation in MPCs

Biologically, KLB acts as a co-receptor for both FGF21 and FGF19. Notably, we found that levels of FGF21, but not FGF19, were significantly higher in plasma from IUGR than NW fetal littermates (Fig. 8A), suggesting that FGF21 may activate KLB to affect muscle development in IUGR fetuses. To investigate this further, MPCs were induced to differentiate in the presence of FGF21. Increasing levels of FGF21 progressively decreased their ability to differentiate into myotubes (Fig. 8B–D), an effect that was associated with a mean decrease in S6K1 phosphorylation (Fig. 8E). These results are consistent with the observed positive effects of KLB down-regulation on myogenesis by MPCs. Finally, FGF21 induced a dose-dependent but not significant ($P > 0.05$) increase in mean KLB expression in MPCs (Fig. 8F).

### FGF21 also reduces the myogenic capacity of human MPCs

We next determined whether FGF21 has the same effects in human and pig MPCs. To do this, we differentiated human fetal MPCs in the presence of increasing concentrations of FGF21. As was the case for pig cells, human cells displayed a decreased ability to undergo myogenesis in the presence of increasing levels of FGF21 (Fig. 9A–C). Moreover, although, on average, mTOR phosphorylation was not affected by FGF21, reduced myogenesis in the presence of FGF21 was associated with a marked decrease in S6K1 phosphorylation (Fig. 8D), indicating that, as in pig cells, FGF21 inhibits mTORC1 signalling in human fetal muscle cells. Finally, FGF21 robustly and dose-dependently stimulated the expression of KLB during myogenic differentiation of human MPCs (Fig. 8E), again highlighting the similarities in responses to FGF21 by human and porcine muscle cells.

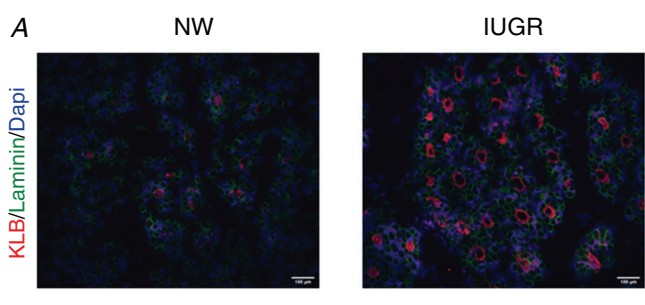

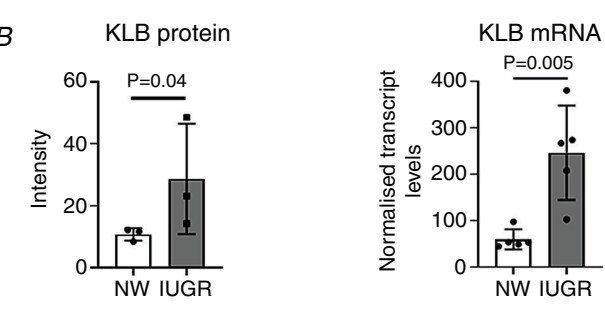

**Figure 5. Relative KLB abundance in skeletal muscle from porcine NW and IUGR fetuses**

*A*, representative cross-sectional images of NW and IUGR semi-tendinosus muscle immunostained for KLB (red). Laminin and DAPI counterstains are shown in green and blue, respectively. Scale bars = 100 $\mu$m. *B*, relative levels of KLB protein (intensity of KLB immunostain) and KLB transcript in NW and IUGR littermates. Mean ± SD values are shown together with individual data points. [Colour figure can be viewed at wileyonlinelibrary.com]

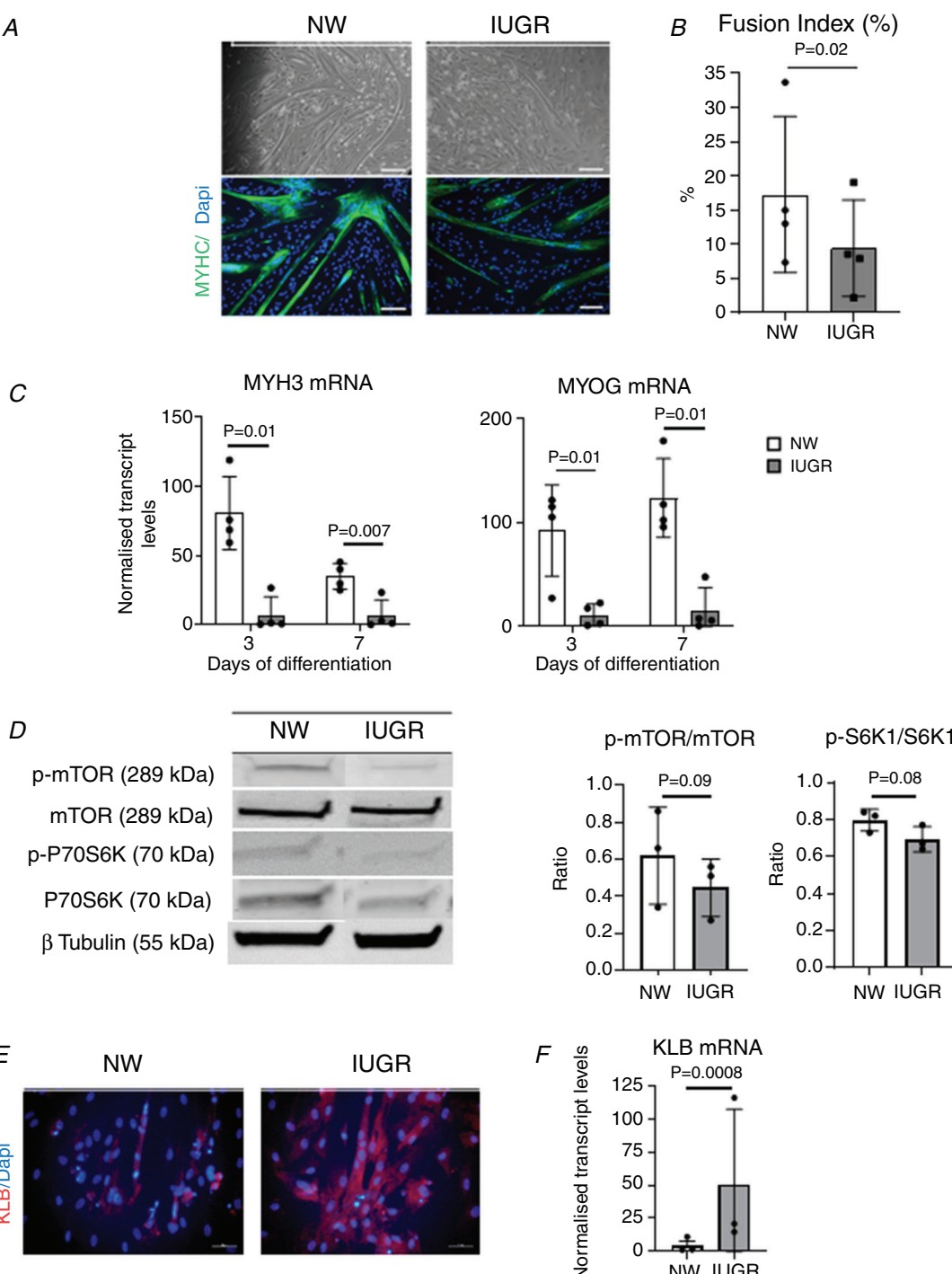

**Figure 6. Effects of IUGR on myogenic differentiation of porcine MPCs**
*A*, representative bright-field (top) and immunofluorescence (bottom) images (scale bars = 100 $\mu$m) of MPCs from NW and IUGR littermates cultured under myogenic conditions for 3 days. MYHC-stained myotubes are shown in green. Nuclear staining with DAPI is shown in blue. *B*, fusion index values obtained from MPC cultures on day 3 of differentiation and calculated from five microscopic fields. *C*, relative levels of MYH3 and MYOG transcripts in MPC cultures on days 3 and 7 of differentiation. Values are shown as fold change expression relative to day 0 values. *D*, representative p-mTOR, mTOR, p-S6K1 and S6K1 immunoblots obtained from MPCs on day 3 of differentiation, together with quantitative values for p-mTOR/mTOR and p-S6K1/S6K1. *E*, representative immunofluorescence images of MPCs on day 3 of differentiation stained with KLB (red); scale bars = 50 $\mu$m. *F*, relative levels of KLB mRNA in MPC cultures from NW and IUGR littermates. Mean ± SD values are shown together with individual data points. [Colour figure can be viewed at wileyonlinelibrary.com]

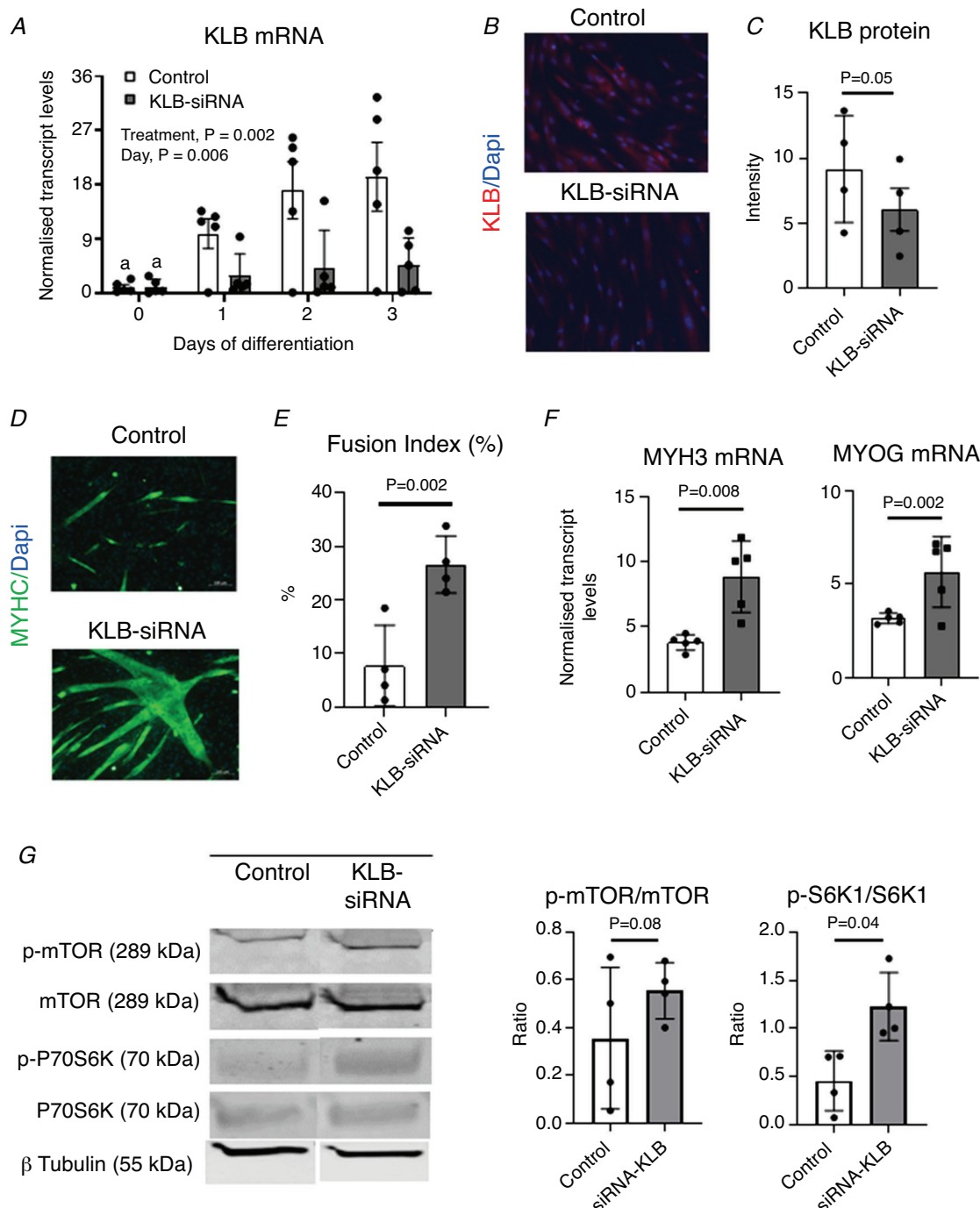

**Figure 7. Effects of KLB down-regulation on differentiation of porcine MPCs**
*A*, relative levels of KLB transcript following transfection of MPCs with KLB siRNA or control siRNA 1 day before myogenic differentiation was induced (day −1). *B*, representative images illustrating KLB immunostaining (red) in MPCs differentiated for 3 days in the presence or absence of KLB siRNA; scale bars = 50 μm. *C*, relative KLB protein levels in the two groups of cells were obtained form quantification of immunostain intensities. *D* to *E*, representative images (*D*) of MYHC immunostaining (green, scale bars = 100 μm) and fusion indices (*E*) in MPCs differentiated for 3 days in the presence or absence of KLB siRNA. *F*, changes in the relative levels of MYH3 and MYOG transcripts in MPCs differentiated for 3 days in the presence or absence of KLB siRNA. *G*, representative p-mTOR, mTOR, p-S6K1 and S6K1 immunoblots obtained from MPCs on day 3 of differentiation, together with quantitative values for p-mTOR/mTOR and p-S6K1/S6K1. Mean ± SD values are shown together with individual data points. [Colour figure can be viewed at wileyonlinelibrary.com]

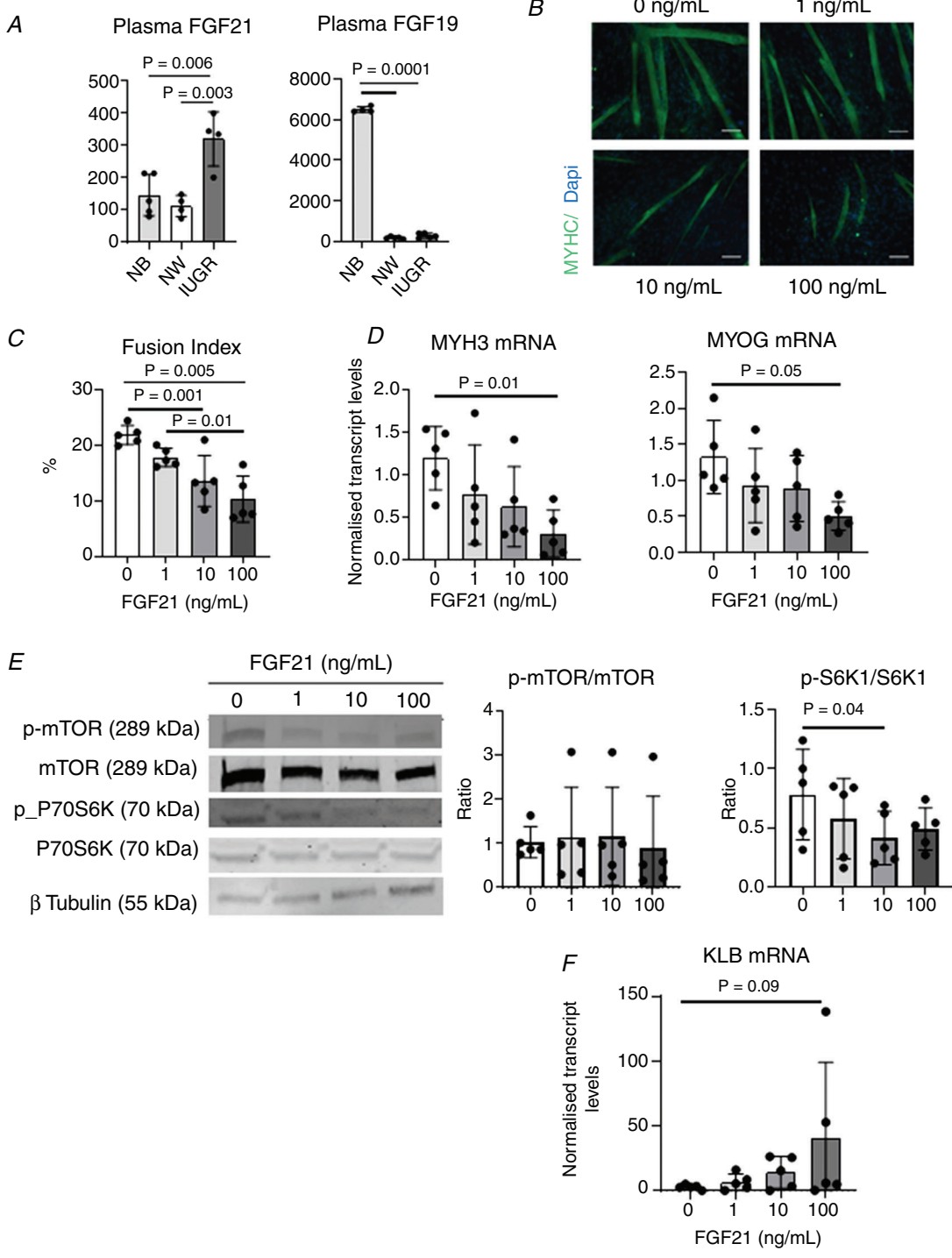

**Figure 8. Effects of stimulation with the KLB agonist, FGF21, on differentiation of porcine MPCs**
*A*, FGF21 and FGF19 levels in the plasma of fetal IUGR and NW littermates, and in healthy newborn piglets (NB). *B*, representative immunofluorescence images of MPCs differentiated for 3 days in the presence of different concentrations of FGF21, as indicated. MYHC staining is shown in green. Scale bars = 100 $\mu$m. *C* to *F*, fusion indices (*C*), relative levels of MYH3 and MYOG transcripts (*D*), representative p-mTOR, mTOR, p-S6K1 and S6K1 immunoblots together with calculated p-mTOR/mTOR and p-S6K1/S6K1 ratios (*E*), and relative levels of KLB transcript (*F*) in MPCs differentiated for 3 days in the presence of different concentrations of FGF21. Mean ± SD values are shown together with individual data points. [Colour figure can be viewed at wileyonlinelibrary.com]

*A*

| 0 ng/mL | 1 ng/mL | 10 ng/mL | 100 ng/mL |

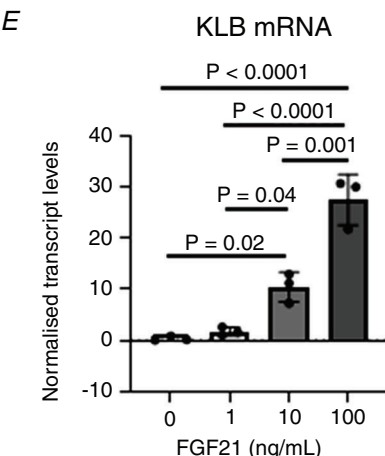

*B* Fusion Index

P = 0.04

*C* MYH3 mRNA

MYOG mRNA

P = 0.005
P = 0.02

*D* FGF21 (ng/mL)

| | 0 | 1 | 10 | 100 |
p-mTOR (289 kDa)
mTOR (289 kDa)
p-P70S6K (70 kDa)
P70S6K (70 kDa)
β Tubulin (55 kDa)

p-mTOR/mTOR

p-S6K1/S6K1

P = 0.008
P = 0.03

*E* KLB mRNA

P < 0.0001
P < 0.0001
P = 0.001
P = 0.04
P = 0.02

**Figure 9. Effects of stimulation with the KLB agonist, FGF21, on differentiation of human MPCs**
*A*, representative immunofluorescence images of MPCs differentiated for 3 days in the presence of different concentrations of FGF21, as indicated. MYHC staining is shown in green. Scale bars = 100 μm. *B* to *E*, fusion indices (*B*), relative levels of MYH3 and MYOG transcripts (*C*), representative p-mTOR, mTOR, p-S6K1 and S6K1 immunoblots together with calculated p-mTOR/mTOR and p-S6K1/S6K1 ratios (*D*), and relative levels of KLB transcript (*E*) in MPCs differentiated for 3 days in the presence of different concentrations of FGF21. Mean ± SD values are shown together with individual data points. [Colour figure can be viewed at wileyonlinelibrary.com]

## Discussion

The present study provides a detailed characterization of genome-wide transcriptional changes in skeletal muscle of the pig IUGR fetus. Widespread activation of tissue injury, in addition to developmental and metabolic pathways, is consistent with reports in tissues from different IUGR models (Vaiman *et al.* 2011; Kelly *et al.* 2017; Rashid *et al.* 2018), and highlights the variety of responses elicited by fetal muscle aimed to protect the developing tissue against hypoxic and other environmental insults. Associated with these were widespread changes in metabolic gene profiles, which were indicative of a switch from glucose to lipids as a source of energy (Limesand *et al.* 2007; Yates *et al.* 2012) and, especially, of increased amino acid catabolism in fetal muscle. In line with the latter observation, previous studies in sheep indicated an increased utilization of muscle protein as a source of body energy to compensate for reduced fetoplacental transport and availability of amino acids (Brown *et al.* 2012; Rozance *et al.* 2018; Stremming *et al.* 2020), an effect that significantly impacts on protein accretion rates and muscle growth in the IUGR fetus. Taken together, the observed changes in metabolic gene profiles in the present study are consistent with the concept that, in response to an imposed deficit of carbohydrates, amino acids (and possibly fatty acids) become a prime source of energy in the IUGR pig fetus, thus contributing to net negative muscle growth. In relation to fatty acids, evidence showing an increase in fatty oxidation in IUGR muscle should be obtained before their role as a primary source of energy in fetal pig IUGR muscle can be definitively established.

A prominent physiological response elicited by a deficiency of amino acids during nutrient restriction is a systemic increase in FGF21 (Solon-Biet *et al.* 2016). FGF21 acts as a master mediator of body-wide responses to starvation, including restricted body growth, aimed at reducing energy expenditure (the so-called 'thrifty phenotype'). In that regard, an increase in the levels of KLB in muscle, together with higher levels of FGF21 in plasma, suggests a key role of this receptor in globally mediating skeletal muscle responses to reduced resource availability in IUGR littermates. Yet, to our knowledge, although FGF21 has been quantified in human term cord blood (Mericq *et al.* 2014), the roles of KLB or FGF21 in developing fetuses with differing growth trajectories have not been reported before. Notably, unlike KLB, our RNA sequencing data did not show differences in the expression of FGFR1 in muscle between IUGR and NW fetuses. KLB pairing with FGFR1 is assumed to account for most of the effects of FGF21 *in vivo* (Kurosu *et al.* 2007); thus, these results indicate that, in the developing fetus, KLB likely acts as the primary regulator of muscle responsiveness to FGF21, consistent with reports in other tissues (Kurosu *et al.* 2007).

To investigate how KLB may mediate the effects of IUGR on skeletal muscle development, we used MPC cultures from both pig and human to validate the significance of our results using the pig as a valuable experimental model for the human. As already indicated, stunted growth, as evidenced by a reduction in the total number and size of myofibres, is the most obvious feature of the IUGR phenotype in skeletal muscle. Using both siRNA-mediated down-regulation of KLB and agonist activation with FGF21 in MPCs, we revealed a causal role of KLB signalling, through inhibition of mTOR, in reduced muscle fibre formation *in vitro*. Studies in genetically-modified mice models showed that muscle-derived FGF21 induced by fasting decreased protein synthesis and increased autophagy, thus reducing total muscle mass (Oost *et al.* 2019). FGF21 is also induced in muscle in response to mitochondrial dysfunction associated with muscle disease or ageing, where a causal link between high FGF21 and muscle mass loss has been established (Tezze *et al.* 2017). Of note, although reduced mitochondrial function is a feature of IUGR (Pendleton *et al.* 2020), we did not detect differences in FGF21 expression in muscle between IUGR and NW littermates, indicating that systemic (presumably derived, at least in part, from fetal liver) rather than local levels of FGF21 may account for its effects in IUGR fetal muscle, at least in the pig. In this context, an effect of FGF21, by mediation of KLB, in restricting muscle growth in IUGR fetuses is consistent with its well-established role with respect to reducing overall body growth as an adaptive energy-saving measure during starvation (Kubicky *et al.* 2012; Wei *et al.* 2012).

KLB is a natural co-receptor of both FGF21 and FGF19, raising the question of whether some of the effects of KLB in fetal muscle may be mediated through binding to FGF19, another endocrine FGF with metabolism-regulatory effects. Indeed, administration of FGF19 induced skeletal muscle hypertrophy and ameliorated muscle wasting in mice (Benoit *et al.* 2017), a finding that is contrary to our conclusion that KLB mediates the inhibitory effects of FGF21 on fetal muscle development. Together with our observation that, unlike FGF21, FGF19 levels in plasma were not different in IUGR and normal fetal littermates and, moreover, were extremely low compared to levels in new-born pigs, this strongly suggests that the observed effects of KLB on IUGR fetal muscle in the present study were mediated by FGF21 rather than FGF19.

Our data indicate that the effects of KLB on fetal IUGR muscle growth are mediated by mTOR. Within mTOR complex 1 (mTORC1), mTOR critically drives muscle growth by stimulating myoblast fusion and protein accretion through phosphorylation of, amongst other targets, S6K1 (Ge & Chen, 2012). Thus, impaired phosphorylation of mTOR and S6K1 was associated

with stunted muscle development in nutrient-restricted cattle and pig fetuses (Zhu *et al.* 2004; Du *et al.* 2005); however, the precise mechanisms involved have not been clarified. Our novel findings provide a valuable step forward towards understanding the mechanisms by which nutritional and metabolic cues affect fetal muscle growth as a result of identifying FGF21-activated KLB as a putative inhibitor of mTORC1 leading to reduced myoblast fusion and muscle fibre growth. Moreover, our findings support the notion that a decrease in mTOR signalling in response to reduced resource availability acts to adjust fetal growth to the capacity of the mother to support fetal needs (Damerill *et al.* 2016; Gupta & Jansson, 2019).

We conclude that adaptation of skeletal muscle to adverse uterine conditions in the porcine growth-restricted fetus involves extensive changes in the activity of cellular pathways involved in tissue growth and development, response to tissue injury, and metabolism. Moreover, results using myogenic cells from pig and human indicate that stimulation of muscle KLB by circulating FGF21 may play a key role in mediating at least some of the adaptive changes to IUGR, most notably a reduction in muscle growth, and that the effects of KLB in muscle cells occur through inhibition of mTOR signalling. Importantly, our results suggest that these effects are conserved in pigs and humans. It must be noted that our conclusions using human fetal cells are based on a relatively small number of biological replicates, and so further studies are warranted to confirm these findings and provide additional mechanistic insight on the effects of IUGR on early human muscle development. The translation of our findings in pigs to humans also needs to take into account that not all cases of IUGR in humans are primarily associated with placental insufficiency (a common feature of IUGR in the pig) and that alternative or additional mechanisms may be involved in disease pathogenesis in those cases. Overall, our results bring new light to the understanding of IUGR pathogenesis in muscle, a developmental adaptation that carries significant risks for life-long health in affected individuals.

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

## Additional information

### Data availability statement

Raw sequencing data files (FASTQ) have been deposited in the NCBI BioProject database (https://www.ncbi.nlm.nih.gov/bioproject/) under accession number PRJNA678714.

### Competing interests

The authors declare that no they have no competing interests.

### Author contributions

The experiments described in this manuscript were carried out at The Roslin Institute, University of Edinburgh. YCA, CJA, CLE and FXD conceived and/or designed the experiments described in this manuscript. All authors were involved in acquisition, analysis or interpretation of the data, as well as in drafting or critical revision of the manuscript for important intellectual content. All authors approved the final version of the manuscript submitted for publication, and agreed to be accountable for all aspects of the work in ensuring that questions related to the accuracy or integrity of any part of the work are appropriately investigated and resolved. All persons designated as authors qualify for authorship, and all those who qualify for authorship are listed.

### Funding

YCA was funded by the National Agency for Research and Development (ANID)/Scholarship Program/DOCTORADO BECAS CHILE/2016 – 72170349. The Roslin Institute receives

funding from The Biotechnology and Biological Sciences Research Council through an Institute Strategic Programme Grant.

## Acknowledgements

We thank the staff at the Dryden Large Animal Unit and Easter Howgate Farm for assistance the with collection of pig samples; James Glover and Anne Saunderson for assistance with human sample collection; and Augustus Donadeu for assistance with image analyses. We are also grateful to Bob Flemming, Graeme Robertson and Barry Bradford at the Roslin Institute for their skilled assistance with tissue processing and bio-imaging.

## Present address

Claire Stenhouse, Physiology of Reproduction, Department of Animal Science, Texas A&M University, College Station, TX, USA

## Keywords

FGF21, fetal, IUGR, KLB, skeletal muscle, transcriptome

## Supporting information

Additional supporting information can be found online in the Supporting Information section at the end of the HTML view of the article. Supporting information files available:

**Peer Review History**
**Table S1**. Primer sequences used for qPCRs.
**Table S2**. Antibodies used.
**Table S3**. List of genes mapped to the porcine genome together with the mean of normalised counts across all samples (base-mean).
**Table S4**. List of genes up-regulated ($P < 0.05$) in muscle from IUGR relative to NW littermates. Genes with FDR $< 0.1$ are shown first. Genes that were validated by qPCR are indicated by a hash symbol (#).
**Table S5**. List of genes down-regulated ($P < 0.05$) in muscle from NW relative to IUGR littermates. Genes with FDR $< 0.1$ are shown first. Genes that were validated by qPCR are indicated by a hash symbol (#).
**Table S6**. List of biological pathways (obtained with Ingenuity Pathway Analyses and grouped by functional category) that were over-represented ($P < 0.01$) among genes differentially expressed (obtained by RNA sequencing) in fetal muscle between IUGR and NW littermates (for details, see Methods)
**Statistical Summary Document**

