## [Peer Review History · The Journal of Physiology]

KLB dysregulation mediates disrupted muscle development in intrauterine growth restriction

Francesc Xavier Donadeu, Yennifer Cortes-Araya, Susan Dan-Jumbo, Claire Stenhouse, Mazdak Salavati, Cheryl J Ashworth, Emily Clark, Cristina Esteves, and William Ho

DOI: 10.1113/JP281647

Corresponding author(s): Francesc Xavier Donadeu (xavier.donadeu@roslin.ed.ac.uk)

Review Timeline:

Submission Date:	17-Jan-2021
Editorial Decision:	15-Mar-2021
Resubmission Received:	20-Nov-2021
Editorial Decision:	14-Dec-2021
Revision Received:	22-Dec-2021
Accepted:	04-Jan-2022

Senior Editor: Laura Bennet

Reviewing Editor: Janna Morrison

Transaction Report:

Dear Dr Donadeu,

Re: JP-RP-2021-281380 "KLB dysregulation mediates disrupted muscle development in intrauterine growth restriction" by Francesc Xavier Donadeu, Yennifer Cortes-Araya, Susan Dan-Jumbo, Claire Stenhouse, Mazdak Salavati, Cheryl J Ashworth, Emily Clark, and Cristina Esteves

Thank you for submitting your manuscript to The Journal of Physiology. It has been assessed by a Reviewing Editor and by 2 Referees and the reports are copied below.

Please let your co-authors know of the following editorial decision as quickly as possible.

As you will see, in its current form, the manuscript is not acceptable for publication in The Journal of Physiology. In comments to me, the Reviewing Editor expressed interest in the potential of this study, but much work still needs to be done (and this may include new experiments) in order to satisfactorily address the concerns raised in the reports.

In view of this interest, I would like to offer you the opportunity to carry out all of the changes requested in full, and to resubmit a new manuscript using the "Submit Special Case Resubmission for JP-RP-2021-281380..." on your homepage.

We cannot, of course, guarantee ultimate acceptance at this stage as the revisions required are substantial. However, we encourage you to consider the requested changes and resubmit your work to us if you are able to complete or address all changes.

A new manuscript would be renumbered and redated, but the original referees would be consulted wherever possible. An additional referee's opinion could be sought, if the Reviewing Editor felt it necessary. A full response to each of the reports should be uploaded with a new version.

I hope that the points raised in the reports will be helpful to you.

Yours sincerely,

Professor Laura Bennet
Senior Editor
The Journal of Physiology
<https://jp.msubmit.net>
<http://jp.physoc.org>
The Physiological Society
Hodgkin Huxley House
30 Farringdon Lane
London, EC1R 3AW
UK
<http://www.physoc.org>
<http://journals.physoc.org>

EDITOR COMMENTS

Reviewing Editor:

Methods Details:

More information about informed consent and food and water to sows.

Comments to the Author:

This is an ambitious study that uses RAN seq to identify genes that may be altered by IUGR in skeletal muscle. As expected for RNA seq analysis, the sample size is low. It is of concern that one sample could not be included. Please be clear about why it was excluded. Eg was it a biological issue or a technical issue. The pig study only used male fetuses and there is a great deal of evidence to suggest that the effect of sex should also be studied. If this could not be done using RNA-seq, it could have been done in the subsequent studies. If there were 6 litters, why were only 4 used in the RNA-seq from the outset? How were they selected?

Were the human samples also from males? Did maternal consent comply with the Helsinki Declaration? Was consent in writing?

Did the sows have access to food and water?

Foetal is misspelt throughout the manuscript.

Line 188 - stable not sable? Were the MIQE guidelines followed?

Was a loading control used in the Western blots? Were the target normalized to the loading control? Please show the blots with the loading controls or the Ponceau S?

Senior Editor:

If the statistical summary document has errors please describe what is incorrect. (Required):

Comments to the Author:

While there is significant interest in your manuscript and exciting findings, there are also significant concerns that the study is significantly underpowered and at this time making a robust scientific conclusion difficult to establish for reviewers. There is concern about total numbers of animals included, and reasons for exclusion as no presentation of sample omission based on analysis and thus that data selection may tend to bias analysis and conclusions reached. In addition to a greater subject n, further details are required regarding technical controls/replication etc.

REFEREE COMMENTS

Referee #1:

This study uses a pig model of naturally occurring IUGR to investigate if muscle tissue from IUGR fetuses have a different transcriptional profile than normal weight fetuses, thereby identifying pathways which may contribute to long term deficits in muscle tissue. This study also uses a range of culturing and gene manipulation techniques to explore the role of key identified pathways. This work is highly original and of interest for the field and does highlight several potential mechanisms which may help to explain long term outcomes previously demonstrated. The study is well written and easy to follow. It must be noted that IUGR is likely a highly heterogeneous developmental complication and that the outcomes of this study likely reflect the perturbed processes that would occur in some but not all affected fetuses. The strength of the conclusions made for this study however are based on a small sample size after removing a sample that would have made the data highly varied and difficult to interpret. I have specific comments relating to these and other concerns below.

1. You highlight that pigs are a species with naturally occurring IUGR. Are the reasons for IUGR in pigs, the same as those in humans (placental insufficiency or other factors) or are they more strongly linked to uterine space restrictions. Can you comment on this in your introduction and reflect on the outcomes you identify and how they may relate/differ to human IUGR?
2. Do you have any data about the degree of reduced oxygen or nutrient availability in the fetuses from which the muscle tissue was taken in this study?.
3. You have selected one normal weight and one IUGR fetus per litter. Can you indicate how many fetuses per litter were IUGR and that average foetal weight did not differ greatly between litters.
4. The sample size used for both the pig study and the human cell experiments are quite low. Although this is understandable given the cost and space requirements for pig research and the difficulties in obtaining human samples, are these sample sizes sufficient based on your original power calculations?
5. Why were only four pairs of the 6 pig samples per group sent for RNA sequencing? Was the litter that was removed from the RNA sequencing analysis different to the other litters in weight, crown rump or any other parameter that could explain the difference? Once, you removed that outlier, did you also remove it from the crown rump and body weight analysis that you have reported? Were the three remaining IUGR fetuses statistically smaller than the control fetuses that were used for RNA seq?
6. It is also concerning that only three human samples were analysed and they varied from 10-20 weeks which would have induced a large degree of variability for these samples. How did these impact your results?
7. Some of tables presented in the manuscript could become supplemental material (tables 2, 3 and 5). I could also not identify table 4 but this would also seem appropriate in supplemental material
8. Why were only male fetuses selected for analysis? Studies have indicated sexual dimorphism in the regulation of IUGR. Thus the data from your study may not reflect what occurs in females. It is now commonly place for both sexes to be

assessed unless a strong reason is provided for not including females.

Referee #2:

In this study the authors identify KLB dysregulation in IUGR piglets using transcriptomic approaches. The experiments go on to establish the role of the KLB (FGF21 co-receptor) in satellite cells isolated from the fetal pig and human skeletal muscle. The data presented provide interesting new information for muscle regulation in general but also potential regulation during fetal growth restriction. A major concern for the experimental approach overall is the small number of samples examined and the lack of information on how these samples were selected from the larger group. Confidence in the results would increase greatly if there was a larger number of animals/subjects evaluated. Also several of the experimental measures, their controls, and replication are not fully described. If these major concerns are addressed, the findings will impact the fields of developmental origins and muscle biology.

Specific comments

- 1) Lines 29 and 45 and 208, change levels to concentrations.
- 2) Line 100 sentence structure/grammar requires attention.
- 3) Line 129 age of gilts at mating and description of the boar used. How were NW males selected if more than 2 were in the litter? Similarly were the fetuses in the IUGR group also selected at random if more were <2 SD. Can litter size per gilt be included? How were the 4 pairs selected for future work (line 266) from the 6 gilts bred and euthanized? Why were only 3 litters used for most of the comparisons? How were they selected?
- 4) The selection of fetuses used in the human cells also is not described where again only an $n=3$ was evaluated. Was sex blocked here as well?
- 5) line 161, Provide RIN scores or measure used to define RNA quality.
- 6) Line 185 What steps were used for primer validation?
- 7) Line 187, how were starting concentrations for the standard curves determined for each specific gene? Where these dilutions used to validate primer efficiencies and not concentrations?
- 8) Line 194, analysis of IHC is missing. Specifically the robustness of the fiber characteristics (fig 1; # fibers evaluated?); KLB expression in Fig 5A (how were intensities measured and normalized?); similar concerns can be expanded to other measurements (Figs 6, 7) where the number of fields or cells counted are not described. Were technical replicates included for cell culture experiments (fig 6-9)?
- 9) Line 206, complete sentence.
- 10) Line 274, What is the rationale for selecting the largest CV to use for the PCA plots? What was the cutoff for excluding litter 4? For the RNA preparation were the 2 NW and 2 IUGR RNA samples combined? If so what step were they mixed? Were there any physiological parameters that set IUGR litter 4 apart? One could argue that litter 3 is also an outlier for IUGRs unless the controls are weighing in on this decision. I would state in line 276 that litter 4 is from the IUGR group.
- 11) Line 271, based on the presentation of the results it appears the very few genes were differentially expressed when FDR was included because p values (not control for multiple comparison) and q (fdr) values <0.1 are used for the functional/pathway analysis. It would be helpful to state you used an approach to maximize gene representation because of the low number of DE genes with $fdr < 0.05$. So to go one step further in defining the DE gene to investigate where factors like fold change considered or low expressing transcripts removed based on a threshold?
- 12) Line 279, the 38 and 43 upregulated genes is confusing because the IUGR were compared to NW. I would not expect genes to change in the control NW group. This includes the presentation of table 5 and 6.
- 13) Figure 4 is confusing. I don't think panel B adds new information when the data is presented in panel A. Also if up and down regulation is relative to NW is presented only one heat map is needed. As presented in the legend on line 759.
- 14) Figure 5A could be presented better the KLB in NW is not well represented.
- 15) Figure 6A the myotubes appear to be greater in the IUGR group, despite lower measures for differentiation.
- 16) I appreciate the focus on using NB cells to evaluate the KLB function but based on the expression difference in IUGR it would be interesting to determine whether siRNA-KLB improved differentiation potential in IUGR MPCs.

17) Line 409, I appreciate the suggestion that fatty acids are utilized as a primary energy source. However, in fetal sheep preparations the oxidation of fat is very low relative to oxygen consumption. Therefore, I would be cautious concluding that there is an increase in FA oxidation in fetal pigs unless evidence is available.

ADDITIONAL FORMATTING REQUIREMENTS:

-Author photo and profile. First (or joint first) authors are asked to provide a short biography (no more than 100 words for one author or 150 words in total for joint first authors) and a portrait photograph. These should be uploaded and clearly labelled with the revised version of the manuscript. See Information for Authors for further details.

-You must start the Methods section with a paragraph headed Ethical Approval. A detailed explanation of journal policy and regulations on animal experimentation is given in Principles and standards for reporting animal experiments in The Journal of Physiology and Experimental Physiology by David Grundy *J Physiol*, 593: 2547-2549. doi:10.1113/JP270818.). A checklist outlining these requirements and detailing the information that must be provided in the paper can be found at: <https://physoc.onlinelibrary.wiley.com/hub/animal-experiments>. Authors should confirm in their Methods section that their experiments were carried out according to the guidelines laid down by their institution's animal welfare committee, and conform to the principles and regulations as described in the Editorial by Grundy (2015). The Methods section must contain details of the anaesthetic regime: anaesthetic used, dose and route of administration and method of killing the experimental animals.

-You must start the Methods section with a paragraph headed Ethical Approval. If experiments were conducted on humans confirmation that informed consent was obtained, preferably in writing, that the studies conformed to the standards set by the latest revision of the Declaration of Helsinki, and that the procedures were approved by a properly constituted ethics committee, which should be named, must be included in the article file. If the research study was registered (clause 35 of the Declaration of Helsinki) the registration database should be indicated, otherwise the lack of registration should be noted as an exception (e.g. The study conformed to the standards set by the Declaration of Helsinki, except for registration in a database.). For further information see: <https://physoc.onlinelibrary.wiley.com/hub/human-experiments>

-Your manuscript must include a complete Additional Information section

-You must upload original, uncropped western blot/gel images (including controls) if they are not included in the manuscript. This is to confirm that no inappropriate, unethical or misleading image manipulation has occurred <https://physoc.onlinelibrary.wiley.com/hub/journal-policies#imagmanip> These should be uploaded as 'Supporting information for review process only'. Please label/highlight the original gels so that we can clearly see which sections/lanes have been used in the manuscript figures.

-A Statistical Summary Document, summarising the statistics presented in the manuscript, is required upon revision. It must be on the Journal's template, which can be downloaded from the link in the Statistical Summary Document section here: https://jp.msubmit.net/cgi-bin/main.plex?form_type=display_requirements#statistics

-Papers must comply with the Statistics Policy https://jp.msubmit.net/cgi-bin/main.plex?form_type=display_requirements#statistics

In summary:

-If $n \leq 30$, all data points must be plotted in the figure in a way that reveals their range and distribution. A bar graph with data points overlaid, a box and whisker plot or a violin plot (preferably with data points included) are acceptable formats.

-If $n > 30$, then the entire raw dataset must be made available either as supporting information, or hosted on a not-for-profit repository e.g. FigShare, with access details provided in the manuscript.

- n clearly defined (e.g. x cells from y slices in z animals) in the Methods. Authors should be mindful of pseudoreplication.

-All relevant 'n' values must be clearly stated in the main text, figures and tables, and the Statistical Summary Document (required upon revision)

-The most appropriate summary statistic (e.g. mean or median and standard deviation) must be used. Standard Error of the Mean (SEM) alone is not permitted.

-Exact p values must be stated. Authors must not use 'greater than' or 'less than'. Exact p values must be stated to three significant figures even when 'no statistical significance' is claimed.

-Statistics Summary Document completed appropriately upon revision

16/11/2021

Dear Editors,

Please find attached revised version of manuscript JP-RP-2021-281380, which has been modified extensively to address all comments and suggestions by both Editors and Reviewers. In particular, as requested, the revised manuscript includes results of new experiments with samples from additional animals. These include qPCR validation results from an additional group of animals to provide added confidence for our RNA sequencing results (new Figure 4), as well as results using cells from new animals (various panels in Figures 5-9) to reinforce our conclusions from in vitro studies. Please note that (as explained in the response to Question 4 of Reviewer 1) after adding the new data some of the differences in Figure 10 in the original manuscript were no longer significant or only tended to significance, therefore we felt our argument about the wider effect of KLB expression was no longer strong enough and we decided to remove that Figure from the revised manuscript. Figure 10 was an add-on to our main body of data, and we believe its omission does not affect the overall conclusions of the study and makes our story stronger.

We have added a new author, William Ho, who generated some of the new data in the revised manuscript

Detailed responses to each comment by the Editors and Reviewers with reference to the specific location in the revised manuscript where each change has been made are below

Many thanks for considering our revised manuscript

F.X. Donadeu (on behalf of all authors)

EDITOR COMMENTS

Reviewing Editor

Methods Details:

More information about informed consent and food and water to sows.

Information on sows (Lines 133-135) and informed consent (Lines 162-166) have been included

Comments to the Author:

This is an ambitious study that uses RAN seq to identify genes that may be altered by IUGR in skeletal muscle. As expected for RNA seq analysis, the sample size is low. It is of concern that one sample could not be included. Please be clear about why it was excluded. Eg was it a biological issue or a technical issue. The pig study only used male fetuses and there is a great deal of evidence to suggest that the effect of sex should also be studied. If this could not be done using RNA-seq, it could have been done in the subsequent studies. If there were 6 litters, why were only 4 used in the RNA-seq from the outset? How were they selected?

Thank you for your comments.

The outlier litter was excluded for subsequent analyses for biological reasons (please see reply to Reviewer 1, Question 5 for details). This has also been mentioned in the revised manuscript (Lines 323-328)

Because most of the IUGR fetuses we obtained were males and we could not predict whether sacrificing more litters would provide the necessary number of female fetuses for a cross-sex study, at the end we were able to get only limited information on females which we decided not to include in the manuscript. A more detailed explanation has been provided in response to Reviewer 1, Comment 8 below.

Justification for using 4 litters for RNA sequencing and how they were selected is provided in the reply to Reviewer 1, Question 5 below

Were the human samples also from males? Did maternal consent comply with the Helsinki Declaration? Was consent in writing?

In relation to the characteristics of the human samples used in our study please see Reply to Reviewer 2, Question 4 below. We can confirm that maternal consent was in writing and that it complied with the Helsinki declaration as indicated in Lines 162-166 of the revised manuscript.

Did the sows have access to food and water?

Yes, they did throughout the study. Information on this has now been included in Lines 133-135

Foetal is misspelt throughout the manuscript.

We have changed foetus/foetal to fetus/fetal throughout the manuscript

Line 188 - stable not sable? Were the MIQE guidelines followed?

Thanks for spotting this. We have changed sable to stable.

New information has been added to Section 2.3 and Table 2 to fully comply with MIQE guidelines.

Was a loading control used in the Western blots? Were the target normalized to the loading control? Please show the blots with the loading controls or the Ponceau S?

Thanks for pointing this out. A loading control (TUBB) was run originally but was not included in the original manuscript. TUBB blots have now been included in the relevant representative images in the revised manuscript (Figures 6D, 7G and 8E), as requested. We quantified the levels of phosphorylated mTOR and S6K1 by normalizing to the level of the respective un-phosphorylated protein. Because we did not quantify absolute mTOR or S6k1 levels in our samples we did not use TUBB for normalization.

Senior Editor

Comments to the Author:

While there is significant interest in your manuscript and exciting findings, there are also significant concerns that the study is significantly underpowered and at this time making a robust

scientific conclusion difficult to establish for reviewers. There is concern about total numbers of animals included, and reasons for exclusion as no presentation of sample omission based on analysis and thus that data selection may tend to bias analysis and conclusions reached. In addition to a greater subject n, further details are required regarding technical controls/replication etc.

Thank you very much for your comments and feedback. To address the comment on low sample numbers, we have now performed new analyses using samples from additional animals, specifically, 1) we have included an additional set of animals in our qPCR validation analyses (Figure 4), and we have included samples from additional animals to our pig cell culture data (various panels in Figures 5 to 8).

We have also provided much more detailed explanation of our experimental procedures both in this document and in the revised manuscript in response to each specific query by the Reviewing Editor and the two Reviewers.

REFeree COMMENTS

Referee #1

This study uses a pig model of naturally occurring IUGR to investigate if muscle tissue from IUGR foetuses have a different transcriptional profile than normal weight foetuses, thereby identifying pathways which may contribute to long term deficits in muscle tissue. This study also uses a range of culturing and gene manipulation techniques to explore the role of key identified pathways. This work is highly original and of interest for the field and does highlight several potential mechanisms which may help to explain long term outcomes previously demonstrated. The study is well written and easy to follow. It must be noted that IUGR is likely a highly heterogeneous developmental complication and that the outcomes of this study likely reflect the perturbed processes that would occur in some but not all affected fetuses. The strength of the conclusions made for this study however are based on a small sample size after removing a sample that would have made the data highly varied and difficult to interpret. I have specific comments relating to these and other concerns below.

Thanks for your comments. Please find reply to specific comments below. All line references are to specific locations in the revised manuscript (unless explicitly indicated) where changes have been made.

1. You highlight that pigs are a species with naturally occurring IUGR. Are the reasons for IUGR in pigs, the same as those in humans (placental insufficiency or other factors) or are they more strongly linked to uterine space restrictions. Can you comment on this in your introduction and reflect on the outcomes you identify and how they may relate/differ to human IUGR?

Thanks for this which is a very valid point. New text has been added in the Introduction (lines 70-74) and Discussion (lines 525-529) to address this.

2. Do you have any data about the degree of reduced oxygen or nutrient availability in the fetuses from which the muscle tissue was taken in this study?.

We did not measure specifically oxygen or nutrient levels in fetal muscle or placenta, however, published data from two of the co-authors (Stenhouse, Ashworth) using the same pig model shows that on Day 90 of gestation there were correlations between fetal weight and each of fetal heart rate, umbilical arterial resistance and the ratio of peak systolic velocity (PSV) to end diastolic velocity (EDV) (Stenhouse et al., 2018, *Reprod Fertil Dev*. 11:1402-1411) and that placenta from the lightest fetuses have impaired angiogenesis (Stenhouse et al.. 2019, *Biology of Reproduction*, 101(1):112–125). These data are consistent with reduced oxygen and nutrient availability to our IUGR fetuses. And indeed it had already been shown that levels of several amino acids were higher in the circulation of average-sized relative to the smallest/IUGR fetus in pig litters during late gestation (Ashworth et al., 2013, *Reprod Fertil Dev*, 25(2):439-45; Lin et al., 2012, *J Nutr.*, 142(6):990-8). We hope that this answers the reviewer's question.

3. You have selected one normal weight and one IUGR fetus per litter. Can you indicate how many fetuses per litter were IUGR and that average fetal weight did not differ greatly between litters.

We realize we should have included this information in the original submission. As requested, the average (\pm SD) number of IUGR fetuses in the different litters as well as litter weights are now indicated in Table 1

4. The sample size used for both the pig study and the human cell experiments are quite low. Although this is understandable given the cost and space requirements for pig research and the difficulties in obtaining human samples, are these sample sizes sufficient based on your original power calculations?

We agree that samples sizes are on the low side however we are confident that with our approach, particularly after the inclusion of additional analyses in this revised version, the conclusions obtained are robust and consistent with previous literature, on which they also build significantly. In particular, doing power calculations for RNA sequencing requires preliminary sequencing data obtained under conditions that closely matched those in the actual experiment. Because this was not available, based on sample sizes reported in previous similar studies (please see references in reply to Comment 5) we decided to use 4 biological replicates. Although we ended up using 3 pairs of animals, results of qPCR validation (Figure 4, now complemented with an additional comparison involving a different set of samples) and immunochemistry (KLB data in Figure 5) provide assurance of the strength of the results obtained with our RNA sequencing approach.

In regard to siRNA and FGF21 data using pig cells, as requested by the Editor and reviewers we performed new experiments and included results from additional animals to provide increased strength to our conclusions. In the case of Figure 10 in the original submission, some of the differences observed there were no longer significant or only tended to significance after adding the new data, therefore we felt our argument about the wider effect of KLB expression was not strong enough and we removed Figure 10 from the manuscript, together with the corresponding text in Results (Lines 375-392 in original manuscript version) and Discussion (Lines 467-478 in the original manuscript version). This was an add-on to our main body of data thus its omission does not bear on the overall conclusions of our study while providing added overall strength to the study.

Given the very limited availability of human foetuses, power calculations from preliminary work with human cells were not possible, and we had to do our experiments with the limited number of samples available (3 foetuses from a total of 4 available). The fact that we detected significant differences that were consistent with those with pig cells provide confidence of our data with human cells. Please see reply to Comment 6 for further clarification.

5. Why were only four pairs of the 6 pig samples per group sent for RNA sequencing?

We submitted a total of 4 litters for sequencing because at the time we were ready to start sequencing analyses we had a total of 4 litters containing a male IUGR, and we decided to go ahead with these samples to a large extent because of the uncertainty associated with sample collection from pig litters, that is, we did not know whether or when we would have any other litters containing at least one fetus that was both IUGR and male (i.e., we could not know this until we had sacrificed each sow and removed all her fetuses). For clarity, we have now mentioned in the Results section that we collected 10 litters but only 8 contained at least one IUGR fetuses, 6 of which male (Lines 309-311). As in several previous similar studies (Tang et al. *Genome Biol.* 2007; 8(6): R115, Zhao et al. *PLoS One.* 2011; 6(5): e19774, Kim et al. *Animal Genetics*; 48: 166-174) which used a limited number of samples (≤ 4 per group), we used RNA sequencing for initial gene screening to be subsequently validated by PCR. The fact that we validated most (all but one) of the differences in gene expression analyzed by PCR and using two different sets of NW littermates provides confidence of the robustness of our data obtained by sequencing.

Was the litter that was removed from the RNA sequencing analysis different to the other litters in weight, crown rump or any other parameter that could explain the difference?

Thanks for raising this which we should have mentioned in our original submission. A distinct feature of the outlier litter is that it was the only one than had two IUGR fetuses (one of each sex), the male fetus used for sequencing being larger than the female IUGR. That is, the outlier was the only litter where the IUGR fetus was not the lightest of ALL fetus in the litter. This may have accounted, at least partially, for the differences in transcriptional profiles relative to the other 3 IUGR fetuses. We have included a sentence to indicate this in Lines 326-328.

Once, you removed that outlier, did you also remove it from the crown rump and body weight analysis that you have reported?

No, the outlier litter was not removed from the Table in the original submission, but we have now excluded that litter from the new Table 1, which includes data only from the litters that we used for analyses

Were the three remaining IUGR fetuses statistically smaller than the control fetuses that were used for RNA seq?

The IUGR fetuses used for sequencing were significantly smaller than control fetuses, even after excluding the outlier litter, as shown in the new Table 1.

6. It is also concerning that only three human samples were analyzed and they varied from 10-20 weeks which would have induced a large degree of variability for these samples. How did these impact your results?

Obtaining human fetal samples for research is extremely challenging, and we're fortunate that procedures are in place at Edinburgh's Medical school to provide such samples. Still, opportunities to harvest human fetal tissues are limited and the developmental stages of available specimens can be quite variable. With this in mind, from all specimens that were made available to us during the project, we collected only those at a relatively late stage (weeks 10-20) which, although not equivalent to a day 90 pig fetus with fully formed secondary fibers, correspond to a stage where secondary fibers are already forming (see for example Barbet et al., 1991, Mechanisms of Development, 35 3 – 11), and which would therefore contain myogenic cells at a developmental stage near those from our pig foetuses. Considering this, we were able to collect samples from a total of four human fetuses, however, we were not able to obtain differentiated myotubes in vitro from one of the specimens. We fully recognize the limitations of having to use only samples from 3 human specimens at slightly different stages of development and the fact that this may limit the power of our analyses. However, the results from all 3 human fetuses were consistent and yielded significant differences which were in agreement with those obtained with pig cells. Yet, some of the differences obtained with human cultures did not reach significance (e.g. mTOR phosphorylation shown in Fig 9D), which warrant further analyses with a large number of samples. Yet, notwithstanding the above limitations, our results clearly show that fetal human cells respond to FGF21 stimulation in a similar manner than those of pig, a finding which we believe is worth reporting and may in the future provide valuable hypotheses for testing related to the pathogenesis of IUGR in human. We have now added text to the Discussion referring to the above-mentioned limitation (Lines 522-525). We hope this alleviates some of the reviewer's concerns.

7. Some of tables presented in the manuscript could become supplemental material (tables 2, 3 and 5). I could also not identify table 4 but this would also seem appropriate in supplemental material

Thanks for the suggestion. Yes, we agree that many of the Tables could be supplemental and we have now included all except Table 1 as supplemental

8. Why were only male fetuses selected for analysis? Studies have indicated sexual dimorphism in the regulation of IUGR. Thus the data from your study may not reflect what occurs in females. It is now commonly place for both sexes to be assessed unless a strong reason is provided for not including females.

This is a very important point. Our decision to focus on males was based on the understanding that, in the pig, males are more sensitive to physiological disruption by IUGR and less likely to catch-up in growth after birth than females (e.g. Gao et al., 2020, BMC Genomics volume 21, Article number 701, and Gonzalez-Bulnes et al., 2012, Reproduction (2012) 144 269–278). Because of this, IUGR males are a substantially bigger production problem for the pork industry than IUGR females. An additional reason for focusing on males only was that out of all litters we sacrificed only 3 contained a female IUGR, one of which together with a male IUGR. We have explained this in Lines 309-311. It is also worth mentioning that in our PCR analyses we originally included samples from litters with female IUGR fetuses, and found that, for all genes shown in Figure 4, expression levels in matched IUGR-normal weight samples were not affected by gender, however, we believe that a larger number of female IUGR-NW pairs may be needed to convincingly rule out gender effects (and we have been unable to collect more litters given the restrictions imposed by the events over the past year) and thus we decided not to include female data in the manuscript. Nonetheless, our data on male IUGR pigs provides a solid story on the mechanisms of IUGR in pigs.

Referee #2

In this study the authors identify KLB dysregulation in IUGR piglets using transcriptomic approaches. The experiments go on to establish the role of the KLB (FGF21 co-receptor) in satellite cells isolated from the fetal pig and human skeletal muscle. The data presented provide interesting new information for muscle regulation in general but also potential regulation during fetal growth restriction. A major concerns for the experimental approach overall is the small number of samples examined and the lack of information on how these samples were selected from the larger group. Confidence in the results would increase greatly if there was a larger number of animals/subjects evaluated. Also several of the experimental measures, their controls, and replication are not fully describe. If these major concerns are addressed, the findings will impact the fields of developmental origins and muscle biology.

Thanks for your valuable comments and suggestions. In response to these and to comments by the Reviewer 1 as well as the Editors, we have now added data from additional animals, and provided a clearer explanation of the experimental approaches used. Below are responses to specific questions with references to the location of changes in the revised manuscript.

Specific comments

1) Lines 29 and 45 and 208, change levels to concentrations.

The change requested has been made.

2) Line 100 sentence structure/grammar requires attention.

The sentence has been re-written to make it clearer.

3) Line 129 age of gilts at mating and description of the boar used.

This information has been added.

How were NW males selected if more than 2 were in the litter?

Thank you. We realize this information was not clear in the original submission. We have now described in detail the criteria used for choosing NW littermates (M&M section, Lines 143-148). We originally collected two different sets of NW littermates, the characteristics of which are shown in the new Table 1. To address the reviewers' concerns on low animal numbers, we have now run qPCR validation analyses using an additional set of NW fetuses (different from that used for sequencing; see Figure 4), which should provide added confidence of the strength of the results of our RNA sequencing analyses.

Similarly were the fetuses in the IUGR group also selected at random if more were <2 SD.

Thank you, this information has also been added. As explained now in Lines 309-311, none of the litters contained more than one IUGR male fetus.

Can litter size per gilt be included?

Average (\pm SD) litter size has been indicated in the footnote in Table 1.

How were the 4 pairs selected for future work (line 266) from the 6 gilts bred and euthanized?

Please see reply to Question 5 by Reviewer 1.

Why were only 3 litters used for most of the comparisons? How were they selected?

We agree that using more than 3 animals per group would strengthen our results and conclusions, and thus following the reviewers' suggestions we have performed new experiments using cells from additional littermate pairs and have added the new data to Figures 5-8 in the revised manuscript. For in vitro experiments, we chose those fetal samples from which we could obtain sufficient number of cells and could be most robustly expanded in vitro.

4) The selection of fetuses used in the human cells also is not describe where again only an n=3 was evaluated. Was sex blocked here as well?

Please see justification for using 3 human fetuses in the reply to Question 6 from Reviewer 1 above. Given the limited availability of human fetuses and the fact that for various reasons obtaining information about fetal sex was not always possible, we were unable to consider gender as a criterion for choosing samples for analyses. Although we fully agree that at least some of the effects of IUGR will be sex-dependent, our purpose using human samples was to establish the effects of KLB knockdown and FGF21 on muscle cells and compare those with effects observed with pig cells, and we believe our approach was suitable for achieving that. We agree that a more detailed examination of IUGR biology in human fetuses would have warranted considering the gender of our human samples, but this was not our objective and would not have been possible in light of restrictions on human sample availability.

5) line 161, Provide RIN scores or measure used to define RNA quality.

Details have been provided as requested (Lines 173-176).

6) Line 185 What steps were used for primer validation?

This has now been detailed in the M&M section (Line 212-215).

7) Line 187, how were starting concentrations for the standard curves determined for each specific gene? Where these dilutions used to validate primer efficiencies and not concentrations?

We have clarified this in the text (Lines 215-217). In particular, a wide range of cDNA dilutions (starting at 1:8 with sequential 4-fold dilutions) were tested during primer validation and this was used to determine the optimum range for subsequent PCRs for each gene. We always use 1:8 as starting dilution as this avoids inhibitory sample effects on polymerase activity observed at lower dilutions.

8) Line 194, analysis of IHC is missing. Specifically the robustness of the fiber characteristics (fig 1; # fibers evaluated?); KLB expression in Fig 5A (how were intensities measured and normalized?); similar concerns can be expanded to other measurements (Figs 6, 7) where the number of fields or

cells counted are not described. Were technical replicates included for cell culture experiments (fig 6-9)?

We apologize for the lack of detail on some of our methods. We have now included details of imaging analyses on tissues (Lines 225-229) and cells (Lines 237-242). In addition, all cell culture experiments were done in triplicate wells and this has now been specified (Lines 265-266). In relation to Figure 6F we noticed that the data in the original KLB protein quantification graph had originated from quantifications of different replicates from a single animal. We have now removed that Figure as we do not think it appropriate to report data from a single biological replicate, and moreover that data provided non-essential information in the context of our study.

9) Line 206, complete sentence.

For simplicity and clarity, the sentence has been incorporated in the new text above.

10) Line 274, What is the rationale for selecting the largest CV to use for the PCA plots?

PC plots should capture the most variation in a dataset, thus the genes with the largest CVs (i.e. including those showing the highest differences among experimental groups) are typically used to generate such plots, and we followed this standard approach.

What was the cutoff for excluding litter 4?

Following visual inspection of the PC plot, litter 4 was confirmed to be an outlier based on the PC2 value for the IUGR sample in that litter being >2 SD away from the mean value of all samples along the PC2 axis. This was confirmed by the fact that the pattern of top differentially expressed genes was different in that litter relative to the other litters.

For the RNA preparation were the 2 NW and 2 IUGR RNA samples combined? If so what step were they mixed?

We understand the reviewer is asking whether RNA samples were combined for the two groups. RNA samples were prepared separately for NW and IUGR from each litter and then submitted for sequencing.

Were there any physiological parameters that set IUGR litter 4 apart?

Yes, there were differences in litter 4 that set it apart from other litters. Please see details in the reply to Question 5 by Reviewer 1.

One could argue that litter 3 is also an outlier for IUGRs unless the controls are weighing in on this decision. I would state in line 276 that litter 4 is from the IUGR group.

Thanks for the suggestion, which we have implemented in Line 325 of the revised manuscript

11) Line 271, based on the presentation of the results it appears the very few genes were differentially expressed when FDR was included because p values (not control for multiple comparison) and q (fdr) values <0.1 are used for the functional/pathway analysis. It would be helpful to state you used an approach to maximize gene representation because of the low number of DE genes with $fdr < 0.05$. So to go one step further in defining the DE gene to investigate where factors like fold change considered or low expressing transcripts removed based on a threshold?

Thanks for the suggestion. Clearer justification for the use of all significantly expressed genes (before FDR adjustment) in IPA analyses has now been provided (Lines 192-195).

Criteria used to select genes for further investigation (PCR validation) have now been included (Lines 359-361).

12) Line 279, the 38 and 43 upregulated genes is confusing because the IUGR were compared to NW. I would not expect genes to change in the control NW group. This includes the presentation of table 5 and 6.

Thanks for spotting this. Genes downregulated in IUGR are now referred as such (rather than as 'upregulated in NW') in both the text and Tables 5 and 6.

13) Figure 4 is a confusing. I don't think panel B adds new information when the data is presented in panel A. Also if up and down regulation is relative to NW is presented only one heat maps is needed. As presented in the legend on line 759.

We concur with the reviewer and have removed panel B from Figure 4.

14) Figure 5A could be presented better the KLB in NW is not well represented.

Thank you for pointing this out. We have now replaced the pictures in Figure 5A with new representative images that clearly show KLB staining in the NW group and also include Laminin staining to make the localization of muscle fibers more obvious.

15) Figure 6A the myotubes appear to be greater in the IUGR group, despite lower measures for differentiation.

Many thanks for spotting this, an obvious mistake from our side that has now been corrected.

16) I appreciate the focus on using NB cells to evaluate the KLB function but based on the expression difference in IUGR it would be interesting to determine whether siRNA-KLB improved differentiation potential in IUGR MPCs.

We absolutely agree with the reviewer, and indeed our original idea was to do siRNA experiments with IUGR muscle cells. However, we found that in general IUGR cells did not grow as robustly as NW cells, and were also more sensitive to the effects of transfection with oligonucleotides, thus we felt we would obtain more robust data using NW cells. Also, since the effects of FGF21 needed to be tested on NW cells, our approach provided consistency across experiments as the same cell type was used for experiments with siRNA and with FGF21. We trust the reviewer will understand the rationale behind our approach.

17) Line 409, I appreciate the suggestion that fatty acids are utilized as a primary energy source. However, in fetal sheep preparations the oxidation of fat is very low relative to oxygen consumption. Therefore, I would be cautious concluding that there is an increase in FA oxidation in fetal pigs unless evidence is available.

Thanks for raising this. We have amended the paragraph to reflect this (Lines 452-454).

Dear Dr Donadeu,

Re: JP-RP-2021-281647X "KLB dysregulation mediates disrupted muscle development in intrauterine growth restriction" by Francesc Xavier Donadeu, Yennifer Cortes-Araya, Susan Dan-Jumbo, Claire Stenhouse, Mazdak Salavati, Cheryl J Ashworth, Emily Clark, Cristina Esteves, and William Ho

Thank you for submitting your manuscript to The Journal of Physiology. It has been assessed by a Reviewing Editor and by 2 expert Referees and I am pleased to tell you that it is considered to be acceptable for publication following satisfactory revision.

The reports are copied at the end of this email. Please address all of the points and incorporate all requested revisions, or explain in your Response to Referees why a change has not been made.

NEW POLICY: In order to improve the transparency of its peer review process The Journal of Physiology publishes online as supporting information the peer review history of all articles accepted for publication. Readers will have access to decision letters, including all Editors' comments and referee reports, for each version of the manuscript and any author responses to peer review comments. Referees can decide whether or not they wish to be named on the peer review history document.

Authors are asked to use The Journal's premium BioRender (<https://biorender.com/>) account to create/redrawn their Abstract Figures. Information on how to access The Journal's premium BioRender account is here: <https://physoc.onlinelibrary.wiley.com/journal/14697793/biorender-access> and authors are expected to use this service. This will enable Authors to download high-resolution versions of their figures.

I hope you will find the comments helpful and have no difficulty returning your revisions within 4 weeks.

Your revised manuscript should be submitted online using the links in Author Tasks Link Not Available.

Any image files uploaded with the previous version are retained on the system. Please ensure you replace or remove all files that have been revised.

REVISION CHECKLIST:

- Article file, including any tables and figure legends, must be in an editable format (eg Word)
- Abstract figure file (see above)
- Statistical Summary Document
- Upload each figure as a separate high quality file
- Upload a full Response to Referees, including a response to any Senior and Reviewing Editor Comments;
- Upload a copy of the manuscript with the changes highlighted.

- A potential 'Cover Art' file for consideration as the Issue's cover image;
- Appropriate Supporting Information (Video, audio or data set https://jp.msubmit.net/cgi-bin/main.plex?form_type=display_requirements#supp).

To create your 'Response to Referees' copy all the reports, including any comments from the Senior and Reviewing Editors, into a Word, or similar, file and respond to each point in colour or CAPITALS and upload this when you submit your revision.

I look forward to receiving your revised submission.

If you have any queries please reply to this email and staff will be happy to assist.

Yours sincerely,

Professor Laura Bennet
Senior Editor
The Journal of Physiology
<https://jp.msubmit.net>
<http://jp.physoc.org>
The Physiological Society
Hodgkin Huxley House
30 Farringdon Lane
London, EC1R 3AW
UK
<http://www.physoc.org>
<http://journals.physoc.org>

REQUIRED ITEMS:

-Please include an Abstract Figure. The Abstract Figure is a piece of artwork designed to give readers an immediate understanding of the research and should summarise the main conclusions. If possible, the image should be easily 'readable' from left to right or top to bottom. It should show the physiological relevance of the manuscript so readers can assess the importance and content of its findings. Abstract Figures should not merely recapitulate other figures in the manuscript. Please try to keep the diagram as simple as possible and without superfluous information that may distract from the main conclusion(s). Abstract Figures must be provided by authors no later than the revised manuscript stage and should be uploaded as a separate file during online submission labelled as File Type 'Abstract Figure'. Please ensure that you include the figure legend in the main article file. All Abstract Figures should be created using BioRender. Authors should use The Journal's premium BioRender account to export high-resolution images. Details on how to use and access the premium account are included as part of this email.

EDITOR COMMENTS

Reviewing Editor:

Thank you for addressing all of the reviewers' comments.

Please remove reference to trends. The statistical test result tells us if there is or is not a difference.

Please include an Abstract figure and legend.

REFEREE COMMENTS

Referee #1:

The authors have undertaken additional analyses to improve the quality of the manuscript. The authors have also addressed each of the major concerns raised. There were several points that were not able to be addressed (such as including females) but in general where possible significant modifications have been made.

Referee #2:

Many of my previous concerns have been adequately addressed in this revision. The additional animal while from the same litters do expand the data set for the different experimental approaches. The additional cell experiments support the role for FGF21 signaling in myogenesis. I have a couple minor suggestions to consider. 1. citation for the statement on line 101-103.

2 line 145. were these NW fetuses (two sets) randomly assigned or were the heavier ones selected for set 1 and lighter for set 2. A brief statement would help the reader understand the groupings.

3. line 176. should it be RIN?

END OF COMMENTS

21/12/2021

Dear Editors,

Please find attached the revised version of manuscript JP-RP-2021-281647X which has been modified to address the remaining comments by the Reviewing Editor and Reviewer 2, as well as to meet the requirements listed in the Revision checklist.

Many thanks for your assistance in getting our manuscript acceptable for publication in J. Physiol

F.X. Donadeu (on behalf of all authors)

Reviewing Editor

References to trends have removed throughout the manuscript (Lines 316,406-409, 422, 434-435 of the revised manuscript)

An abstract figure and legend (Lines 36-47) have been included

Referee #2

The citation requested has been added (Lines 116-117)

Clarification has been provided on the NW groupings used (Lines 169-172)

RINe (rather than RIN) is correct and it stands for RIN equivalent (as spelled out in Line 193)

Dear Dr Donadeu,

Re: JP-RP-2021-281647XR1 "KLB dysregulation mediates disrupted muscle development in intrauterine growth restriction" by Francesc Xavier Donadeu, Yennifer Cortes-Araya, Susan Dan-Jumbo, Claire Stenhouse, Mazdak Salavati, Cheryl J Ashworth, Emily Clark, Cristina Esteves, and William Ho

I am pleased to tell you that your paper has been accepted for publication in The Journal of Physiology.

IMPORTANT

We have a note that we still need a high resolution 'BioRendered' figure from you. Diana will be in touch about this.

NEW POLICY: In order to improve the transparency of its peer review process The Journal of Physiology publishes online as supporting information the peer review history of all articles accepted for publication. Readers will have access to decision letters, including all Editors' comments and referee reports, for each version of the manuscript and any author responses to peer review comments. Referees can decide whether or not they wish to be named on the peer review history document.

Are you on Twitter? Once your paper is online, why not share your achievement with your followers. Please tag The Journal (@jphysiol) in any tweets and we will share your accepted paper with our 23,000+ followers!

The last Word version of the paper submitted will be used by the Production Editors to prepare your proof. When this is ready you will receive an email containing a link to Wiley's Online Proofing System. The proof should be checked and corrected as quickly as possible.

Authors should note that it is too late at this point to offer corrections prior to proofing. The accepted version will be published online, ahead of the copy edited and typeset version being made available. Major corrections at proof stage, such as changes to figures, will be referred to the Reviewing Editor for approval before they can be incorporated. Only minor changes, such as to style and consistency, should be made a proof stage. Changes that need to be made after proof stage will usually require a formal correction notice.

All queries at proof stage should be sent to TJP@wiley.com

Yours sincerely,

Professor Laura Bennet
Senior Editor
The Journal of Physiology
<https://jp.msubmit.net>
<http://jp.physoc.org>
The Physiological Society
Hodgkin Huxley House
30 Farringdon Lane
London, EC1R 3AW
UK
<http://www.physoc.org>
<http://journals.physoc.org>

P.S. - You can help your research get the attention it deserves! Check out Wiley's free Promotion Guide for best-practice recommendations for promoting your work at www.wileyauthors.com/eoo/guide. And learn more about Wiley Editing Services which offers professional video, design, and writing services to create shareable video abstracts, infographics, conference posters, lay summaries, and research news stories for your research at www.wileyauthors.com/eoo/promotion.

* IMPORTANT NOTICE ABOUT OPEN ACCESS *

Information about Open Access policies can be found here <https://physoc.onlinelibrary.wiley.com/hub/access-policies>

To assist authors whose funding agencies mandate public access to published research findings sooner than 12 months after publication The Journal of Physiology allows authors to pay an open access (OA) fee to have their papers made freely available immediately on publication.

You will receive an email from Wiley with details on how to register or log-in to Wiley Authors Services where you will be able to place an OnlineOpen order.

You can check if your funder or institution has a Wiley Open Access Account here <https://authorservices.wiley.com/author-resources/Journal-Authors/licensing-and-open-access/open-access/author-compliance-tool.html>

Your article will be made Open Access upon publication, or as soon as payment is received.

If you wish to put your paper on an OA website such as PMC or UKPMC or your institutional repository within 12 months of publication you must pay the open access fee, which covers the cost of publication.

OnlineOpen articles are deposited in PubMed Central (PMC) and PMC mirror sites. Authors of OnlineOpen articles are permitted to post the final, published PDF of their article on a website, institutional repository, or other free public server, immediately on publication.

Note to NIH-funded authors: The Journal of Physiology is published on PMC 12 months after publication, NIH-funded authors DO NOT NEED to pay to publish and DO NOT NEED to post their accepted papers on PMC.

EDITOR COMMENTS

Reviewing Editor:

Thank you for revising the paper.

REFEREE COMMENTS

Referee #2:

The authors have adequately addressed my minor suggestions.

2nd Confidential Review

22-Dec-2021